

# Review of Radar Altimetry Techniques over the Arctic Ocean: Recent Progress and Future Opportunities for Sea Level and Sea Ice Research

Graham D. Quartly[1], Eero Rinne[2], Marcello Passaro[3], Ole B. Andersen[4], Salvatore Dinardo[5], Sara Fleury[6], Kevin Guerreiro[6], Amandine Guillot[7], Stefan Hendricks[8], Andrey A. Kurekin[1], Felix L. Müller[3], Robert Ricker[8], Henriette Skourup[4], and Michel Tsamados[9]

[1]Plymouth Marine Laboratory, Plymouth, PL1 3DH, UK
[2]Finnish Meteorological Institute, Erik Palménin aukio 1, FI-00560 Helsinki, Finland
[3]Deutsches Geodätisches Forschungsinstitut, Technische Universität München (DGFI-TUM), München, Germany
[4]DTU Space, National Space Institute, Elektrovej Bygning 327, 2800 Kongens Lyngby, Denmark
[5]He Space, Robert-Bosch-Strasse 7, 64293 Darmstadt, Germany
[6]LEGOS, 4 Avenue Edouard Belin, 31400 Toulouse, France
[7]Centre National d'Etudes Spatiales (CNES), 18 Avenue Edouard Belin, 31400 Toulouse, France
[8]Alfred Wegener Institute, Am Handelshafen 12, 27570 Bremerhaven, Germany
[9]Centre for Polar Observation and Modelling, Earth Sciences, University College London, London, UK

**Correspondence:** G.D. Quartly (gqu@pml.ac.uk)

**Abstract.** There are numerous needs for monitoring sea level and sea ice in the Arctic, ranging from concern about changes in ice cover being both an indicator and a driver of long-term climate change to shipping interest in alternative routes and the associated risks to the safety of vessels and crew. Furthermore, sea level relative to the geoid allows us to quantify the geostrophic circulation, including any changes in the flow. Radar altimeters provide an important means of quantifying changes

in sea level and sea-ice thickness, although there are increased complexities in the interpretation of their data over such a variable surface. This paper reviews the techniques for deriving useful geophysical information over a mix of leads and ice floes, covering the approaches for both conventional (low rate mode) altimetry and the newer delay-Doppler (Synthetic Aperture Radar) instruments. It discusses the challenges in discriminating the returns from different surfaces, the retracking approaches and the corrections required. The review finishes with a look ahead to how new technologies, analyses and understanding may

be expected to improve the monitoring in this critical environment.

## 1 Introduction

Altimetry is a standard geophysical technique for measuring Earth surface properties at the sub-satellite point, chief of which is the range, which when combined with predicted height of the satellite and a host of geophysical correction terms gives the elevation of the reflecting point. Such information is used to monitor sea level rise, geostrophic currents in the ocean,

lake and river levels, and evolution of the ice sheets (Fu and Cazenave, 2001). A useful summary of the key Arctic science results derived from altimetry is provided by Johannessen and Andersen (2017). This paper reviews the technical developments





within altimetry in the context of the marine sector of the Arctic (Figure 1), which has a number of particular issues related to the spatial and temporal coverage, and the need to distinguish between different-natured reflecting surfaces and their distinct corrections. The key parameters to be determined are the sea level and the freeboard (i.e. the height of the sea-ice surface above the expected sea level), from which sea-ice thickness can be estimated. First we consider the scientific and economic

imperatives that necessitate altimetric monitoring of the Arctic, including the temporal and spatial scales that would ideally be desired for those applications.

## 1.1  Scientific and operational requirements

The Arctic environment is changing dramatically, such that over the past 50 years the Arctic has warmed more than twice as rapidly as the rest of the Earth (AMAP, 2017). In February this year (2018), the north of Greenland has been around 15°C

warmer than average for this month, and sea temperature is also increasing, which contributes to the global sea level rise by thermal expansion. Other indicators of the changes occurring in the Arctic are the numerous records for lowest sea-ice extent and thickness during the past years. Despite strong interannual variability, sea-ice extent clearly continues a long-term downward trend. By extrapolating the recent observations, we can predict an ice-free Arctic Ocean by the late 2030s (Screen and Williamson, 2017; Jahn, 2018), which is earlier than previously expected. These long-term changes affect the Arctic

ecosystems, and also the weather in lower latitudes (e.g. extreme cold events in Europe, Southeast Asian monsoons), due to the modifications of atmospheric circulation (Rinke et al., 2006).

To understand these issues of climate change, a number of different Earth Observation (EO) datasets are brought together: sea-ice area, concentration, drift, thickness as well as the topography of the ocean at global and regional scales. In climate change studies based on satellite data, it is a major challenge to construct homogeneous time series from consecutive satellite

sensors, which are what are required for the detection of changes over several decades. This is the main goal of the Climate Change Initiative from ESA, with projects both on Sea Level (Quartly et al., 2017; Legeais et al., 2018) and Sea Ice (Paul et al., 2018).

On the other hand, operational applications today require accurate products over the Arctic with minimal time delay. Modelers need Arctic sea level and sea-ice thickness products on a daily basis. Navigational needs include products detailing ice-free

time and zones, ice volume, ice drift, snow on sea ice, particularly for data along the main intended shipping routes, where the requirement is for ice concentration, channel portion (open water portion that can be used for navigation), ridging rates and sizes for different ice types. Daily evolutions and high spatial resolution are of great interest, but ship operators also need long time series of sea-ice thickness when planning operations in the Arctic. The Polar Code International Maritime Organization (2017) requires manuals for ship operations to include considerations of ice conditions in their planned area of operations.

Altimeter sea-ice thickness estimates can be used to complement other sources of information in this planning.

Improving our ability to project future sea level rise and sea-ice conditions, implies developments in both observing systems (to constrain and validate the models with present and recent past observations) and in modelling of various processes at different spatial and temporal scales. Although significant progress has been made in the past decade, there are significant





**Figure 1.** a) Bathymetry of the Arctic showing main features and circulation (BG - Beaufort Gyre; TPD - Transpolar Drift; EGC - East Greenland Current). b) Seasonal variation of ice cover (white indicates region with median sea-ice concentration from NSIDC (Cavalieri et al.) > 50% in September; light blue the extent of 50% cover in March, and dark blue generally open ocean). Red circles denote limit of altimeter coverage (81.5° for ERS-1/ERS-2/Envisat/AltiKa; 88° for CryoSat-2). Grey indicates the area not monitored by wide-swath passive microwave sensors such as SSM/I.



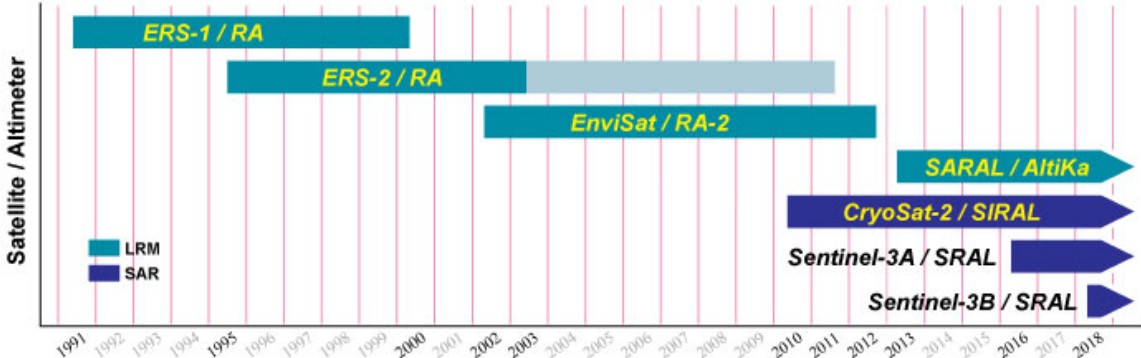

**Figure 2.** Past and current radar altimeter missions with an orbit poleward of 72°N and separated into traditional low-resolution measurement (LRM) and synthetic aperture radar (SAR). Years of mission launch and end are highlighted; ERS-2 only provided limited data after June 2003.

benefits in improving all the observing systems needed to measure and interpret sea-ice characterization and sea level change as well as improving the modelling of future projections of ice melt and global sea level rise and their regional impacts

## 1.2 Relevant altimeter datasets

Altimeters provide nadir measurements of range, with a fine along-track spacing (typically every 300 m), but with no swath coverage. This means that with a single altimeter, tens of days of along-track data are required in order to give good spatial sampling, and simply to provide homogeneous maps of sea level on a monthly basis requires analysis and homogenization of the data from multiple altimeters (see Figure 1 of Quartly et al. (2017)). For Arctic analyses, there is the complication that whilst there is dense coverage at the satellite's turning latitude (defined by its orbital inclination), there is no coverage beyond that. Since the chosen orbit for a satellite is a compromise with other mission demands, only a few altimeters have provided data north of 72°N (the limit for Geosat and Geosat Follow On).

The European Space Agency (ESA) has launched a series of satellites (see Figure 2 and Table 1) into orbits with an inclination of 98.5°, enabling coverage up to 81.5°N. These were ERS-1 (launched in 1991), ERS-2 (1996) and Envisat (2002), all with operation at Ku-band (13.6 GHz), with the principal differences being that Envisat had a higher pulse repetition frequency (PRF) enabling more independent pulses to be averaged in each 0.05 seconds, and also had an additional frequency of operation, in S-band (3.2 GHz).

The same orbit was adopted by SARAL/AltiKa, which is the only altimeter to eschew Ku-band in favour of Ka-band for operations (Steunou et al., 2015); this gives it a slightly smaller ground footprint than other LRM (low-resolution measurement) sensors, and less sensitivity to delays by free electrons in the ionosphere. Radio waves at Ka-band are much more affected by moisture in the troposphere than is the case for Ku-band, but, this is not relevant for polar studies. The other radar altimeters to have covered the Arctic are CryoSat-2, Sentinel-3A, and the recently-launched Sentinel-3B, all of which operate in delay-Doppler (Synthetic Aperture Radar) mode. This latter mode of operations enables finer along-track resolution and



**Table 1.** Performances characteristics of the altimeters covering the Arctic Ocean

| Satellite / Altimeter | Dates | Radar freq. (GHz) | Band-width (MHz) | Beam-width (°) | Repeat period (days) | Comments |
|---|---|---|---|---|---|---|
| ERS-1 / RA | Jul 1991 - Apr 2000 | 13.8 | 330 | 1.3 | 35 | There were also mission phases with 3-day and 168-day repeat orbits. |
| ERS-2 / RA | Apr 1995 - Jun 2003 | 13.8 | 330 | 1.3 | 35 | Onboard data storage lost in June 2003; limited direct-relay data available until July 2011. |
| Envisat / RA-2 | Mar 2002 - Apr 2012 | 13.6 3.2 | 320 160 | 1.3 5.5 | 35 | Operation in S-band ceased in Mar. 2008. Changed to 27-day repeat orbit in Oct. 2010. |
| SARAL / AltiKa | Feb 2013 - | 35 | 100 | 0.6 | 35 | Drift orbit from Jul 2016 onwards. |
| CryoSat-2 / SIRAL | Apr 2010 - | 13.6 | 320 | 1.08 x 1.2 | 369 | Non-circular antenna. Operated in SAR mode over sea ice. |
| Sentinel-3A / SRAL | Feb 2016 - | 13.6 5.4 | 350 320 | 1.35 3.4 | 27 | |
| Sentinel-3B / SRAL | Apr 2018 - | 13.6 5.4 | 350 320 | 1.35 3.4 | 27 | |

The first 4 rows detail LRM instruments, with a circular pulse-limited footprint in low wave height conditions of ∼2-3 km diameter (Chelton et al., 1989).The last 3 rows concern SAR altimeters, for which the Doppler processing reduces the along-track resolution to ∼300 m. For a given altitude orbit, a satellite with a long revisit time provides a finer network of tracks.

lower noise levels (due to the superposition of multiple viewing geometries). The resultant radar echoes ("waveforms") have a more impulse-like shape (see Figure 3). Although their output can be degraded into a pseudo-LRM waveform and processed similarly to previous LRM instruments, it is preferable to develop new processing techniques to utilise the greater information content available in these SAR waveforms.

5     In the succeeding sections we discuss first how these waveform data and derived fields are used to discriminate between different reflecting surfaces (principally open ocean, leads and floes), and then how these data are subsequently retracked i.e. processed to give an accurate estimate of range.





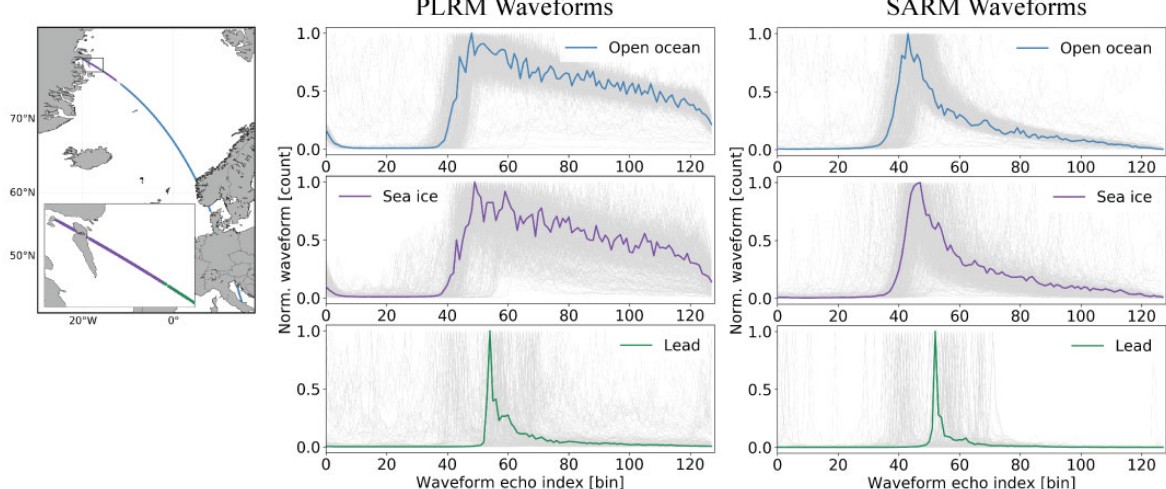

**Figure 3.** Distinction between altimeter reflections from open ocean, sea-ice floes and leads. Data are from a single track of Sentinel-3A, processed as both Pseudo-LRM (middle column) and SAR mode (right-hand column). The coloured lines show a single specimen waveform, whilst the cloud shows the variation in position and shape within this portion of track approaching the east coast of Greenland. [Image courtesy of Ben Loveday, PML.]

## 2 Waveform discrimination

It is important to be able to distinguish accurately between reflections from different surfaces (open ocean, floes and leads) as different stages of processing are usually applied to each surface type. This section first reviews the methodologies available to achieve this discrimination.

### 2.1 LRM altimeters

Reflections from open ocean normally produce waveforms to which a smooth curve can be fitted (see Fig. 4a. For LRM echoes, the leading edge width (LEW) is related to the wave height, and the normalized backscatter, $\sigma^0$, to the surface roughness (Brown, 1977; Hayne, 1980), and the trailing edge slope (TES) sometimes associated with mispointing of the satellite (Amarouche et al., 2004). The parameters LEW, TES and $\sigma^0$ are also relevant over continuous ice floes, although the waveform shape may be slightly different (Figure 3) and the reflectivity of ice will give a different range of $\sigma^0$ values. LRM waveforms over leads in the sea ice have a significantly different shape, because the calm conditions within the lead provide a mirror-like surface and thus high backscatter values. The power in the signal is then confined to a small number of waveform bins.





**Figure 4.** Example radar waveforms for ocean, leads and ice floes and the parameters derived from them. a) Schematic of a fitted LRM waveform (suitable for ocean or ice floes), showing the characteristics identified by Leading Edge Width (LEW), Trailing Edge Slope (TES), amplitude ($\sigma^0$) and pulse peakiness ($PP$), where $P_{max}$ and $P_{mean}$ are from the actual waveform rather than the smooth fitted shape. b) Example LRM waveforms from Envisat, also showing the hard limit on counts for AltiKa waveforms. c) and f) Example stacks of range-corrected SAR data from CryoSat-2 for ice floe and lead respectively. d) Range-integrated power for both example stacks (note different axes), plus illustration of fitting of Stack Standard Deviation (SSD) and calculation of Stack Peakiness (SP). e) Integrated SAR waveforms for ice floe and lead (note different axes).





### 2.1.1 Classical techniques

The simplest approaches to surface discrimination identify certain waveform characteristics which have significantly different values over disparate surfaces, and use permitted ranges to assign the data to different classifications. Several of the parameters used are descriptors of the fitted waveform e.g. LEW, TES and $\sigma^0$, which are shown in Figure 4a. Various definitions of $\sigma^0$ may exist within the data stream according to which model is fitted. Another characteristic of the waveform is pulse peakiness ($PP$), which was introduced by Laxon (1994b), and is defined as the ratio of the peak power in any of the waveform bins to the mean power over those bins expected to hold the signal (i.e. from the leading edge onwards). There are variants as some users chose different integration ranges or normalization factors; but a generally-accepted implementation is included in modern GDRs (Geophysical Data Records). Zakharova et al. (2015) note that there was a hard limit on the waveform counts for AltiKa of 1250 (see Figure 4b), affecting 6-10% of data. This limited the usefulness of $PP$ for lead discrimination for that mission, especially as different settings for the AGC (Automatic Gain Control) would change the scaling of the waveform, and thus the mean number of counts, but not the maximum. Therefore for surface discrimination for AltiKa they introduced a new term, maximal waveform power ($MP$):

$$MP = 10log_{10}(max(P_i)) + AGC \qquad (1)$$

where $P_i$ is the power level in the $i^{th}$ waveform bin.

Although there are basic physical models governing the expected waveform shape over open ocean and ice floes, the precise thresholds used in the classification tend to be chosen empirically by various research teams. Usually the actual thresholds are not important; what limits the efficacy of the classification is whether the assumptions behind the evaluation are valid. Indeed the distribution of values for PP depends upon the waveform shape (and thus on radar frequency) and on the number of pulses summed to generate an "average waveform". The sea-ice processing chain at University College London (UCL) has used $PP$ and $\sigma^0$, with additional contextual information from daily composites of sea ice concentration (SIC) from passive microwave satellites, and from this generated the first pan-Arctic view of sea-ice thickness from ERS-1 and ERS-2 (Laxon et al., 2003) and the first accurate mean sea surface for the Arctic (Peacock and Laxon, 2004). The "multiple criteria" approach used by Poisson et al. (2018) also included some constraints on $PP$, $LEW$ and $\sigma^0$. For AltiKa, Zakharova et al. (2015) used $MP$, which is akin to a combination of peakiness and $\sigma^0$ (as the AGC setting is the altimeter's delayed on-board response to changes in $\sigma^0$).

### 2.1.2 Statistical techniques

In the preceding section the waveform classification was based on simple thresholds for various criteria, with those bounds set by the user on the basis of physical insight and empirical interpretation of histograms of those parameters. Various machine-learning approaches have been developed that utilise training datasets to improve on those classical methods; two of these are described here.



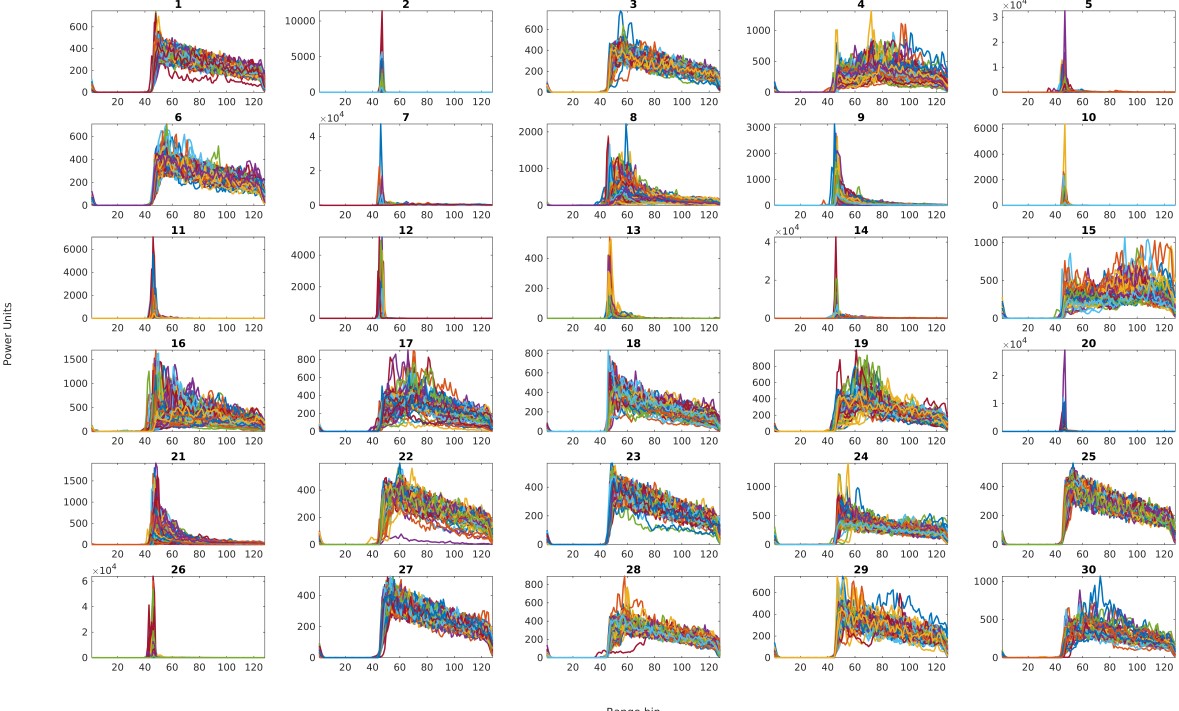

**Figure 5.** Envisat waveform clusters after K-medoids clustering showing selected waveforms (every twenty fifth per cluster). [Image from Müller et al. (2017).]

Müller et al. (2017) produced a clustering algorithm that used six waveform features in an unsupervised classification (i.e. a system in which there no pre-defined target groupings). For this they needed a reference dataset comprising the majority of all possible scatter types. The partitioning of data is achieved using the K-medoids clustering algorithm, which categorizes waveforms into a pre-defined number of K classes. Figure 5 shows the separation of more than 300000 Envisat waveforms
into 30 classes. These clusters are then assigned to different surface conditions in order to condense them to "ocean", "ice floe" and "lead/polynya" returns. This is achieved by analysing the mean feature values of each cluster and comparing selected exemplary waveforms with SAR data (see Sect. 2.3.2). For example, clusters displaying very narrow and clear single-peaked waveforms (e.g. cluster numbers 2, 10, 11, 12, 20 and 26) are assigned to the lead/polynya clusters. Waveforms exhibiting a typical ocean-like shape, characterized amongst other aspects by a weak trailing edge slope or decreased maximum power
value, (e.g. clusters 1, 3, 6 and 25) are allocated to ocean waveforms. Remaining clusters represent ice returns. If there is no clearly interpretable signature, then they are set to "undefined" (e.g., clusters 5 and 15). Afterwards, the obtained waveform model is used to classify and label all remaining waveforms. This is done by K-nearest neighbour, which is a memory-based classifier method.

    Another possible classification method is based on the use of a Neural Network (NN), as described in Gommenginger et al.
(2011) and Poisson et al. (2018). This approach is a supervised classification i.e. the neural net is trained to associate particular



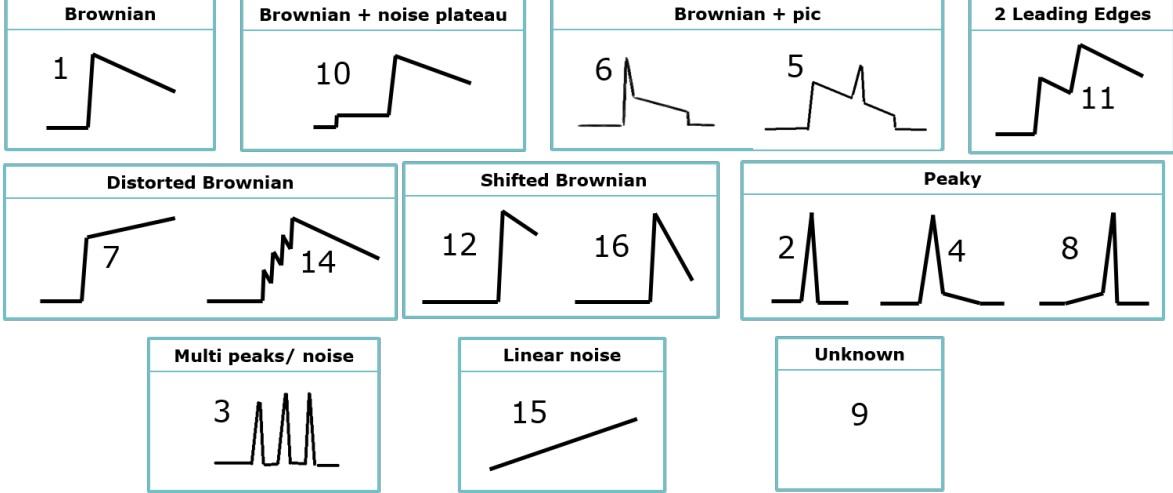

**Figure 6.** Cartoons of the 15 different shapes used for neural net classification for both Envisat and AltiKa waveform data. [Image courtesy of Jean-Christophe Poisson, CLS.]

waveform shapes with user-specified classes (Figure 6). Again the input data is not the full set of waveform bins, but a small set of characteristics describing the shape and amplitude of the waveform, which include $PP$, TES and LEW, as well as the existence or not of extra peaks in the trailing edge. Neural network classifiers directly model discriminant functions (functions enabling the prediction of group membership of a sample based on the values of the input predictive variables) if the output

values are defined in an appropriate way. It requires a large training dataset with enough representative examples of each of the predefined classes.

The purpose is to classify the different geometrical shapes of the echoes (see Fig. 6) and not the different surfaces, even if some links can be made between them. It is important to define not only classes for all echo shapes of interest, but also for all other waveforms numerous enough to impact the classifier. Even if they do not provide useful information, their identification

as a dedicated class number prevents the algorithm from misclassifying them as shapes of interest. Poisson et al. (2018) used 12 classes to cover all possible Envisat waveforms in the Arctic, with one of those corresponding to "ocean", one to "leads" and three of them subsequently amalgamated to specify "floes". The remaining classes were deemed too complex for further interpretation and retracking. This method has been expanded to also work with waveforms from AltiKa and Sentinel-3 (Longépé et al., 2018).

**2.2    SAR altimeters**

In SAR altimetry, both the time delay and the Doppler shift of the echoes are recorded. Whilst the time delay indicates, as for LRM, which annulus about the nadir point is contributing, the Doppler shift gives the position fore or aft of the satellite flight direction; together these give a much finer resolution cell (Raney, 1998). A SAR altimeter thus provides multi-look viewing for each sub-satellite point; range correction then aligns these multiple records for a given point within a "waveform stack" (Fig.



4c,f). An incoherent sum over all look directions gives a SAR waveform (Fig. 4e), which is sharper than an LRM waveform because of the finer footprint achieved through Doppler processing, and has a lower noise level due to the higher number of pulses averaged. Alternatively the stack may be summed in the orthogonal direction to give the Range Integrated Power (RIP, see Fig. 4d).

Both waveforms and RIP will be peaky when the satellite moves over a very smooth surface, such as a lead. For open sea and ice, i.e. for areas in which the scatterers have different orientations (diffuse scattering of rough surfaces), the RIP will be closer to a Gaussian shape, with a much larger value for the SSD (Stack Standard Deviation). The Stack Skewness (SS) and Stack Kurtosis (SK) are other characteristics calculated from the RIP and provided within the GDRs. Because of its finer along-track resolution, SAR altimetry therefore has the potential to identify smaller leads. Nevertheless, waveforms returned over a floe

may still be dominated by bright reflections from off-nadir leads.

The current research on SAR waveform discrimination can be divided into three main techniques, which are described below.

### 2.2.1   Power-based methods

Power-based methods build on the assumption that the smoother the illuminated surface is at nadir, the higher will be the power received back to the altimeter. This not only works for the discrimination between sea ice and leads, but also off-nadir

leads are usually characterised by lower $\sigma^0$ values compared with leads at nadir (Wernecke and Kaleschke, 2015). However the determination of the appropriate threshold, in particular to avoid off-nadir leads, is challenging. The absolute value of the returned power is affected by the proportion of sea ice in the illuminated area and its characteristics, the size of leads, and the presence of refrozen areas within leads. A conservative choice of a high threshold minimizes the false detection of leads, but considerably reduces the number of leads detected and thus may provide insufficient coverage of sea level data. Indeed, it may

be necessary to use different threshold values in different regions (Passaro et al., 2017). Passaro et al. (2017) advocate using a relative power i.e. the ratio between the maximal power for a given waveform and the median value for that region.

### 2.2.2   RIP and waveform shape-based methods

Various stack characteristics based on either the RIP or the SAR waveform have been proposed to help with the classification, especially in the discrimination between echoes from leads at nadir and off-nadir. In the original specification for the CryoSat-2

processing , Wingham et al. (2006) had already detailed that a Gaussian shape be fitted to the RIP, with the SSD, SS and SK all being automatically determined. Passaro et al. (2017) have also defined a stack peakiness (SP) in a similar manner to pulse peakiness; this is a characteristic that is independent of the fitting of a Gaussian. An alternative approach to elucidating the off-nadir leads has been to define "left" and "right" PP values, corresponding to the maximum power divided by the mean power over just those waveform bins immediately before or after the peak Ricker et al. (2016). Figure 7 shows that all the indicators

show strong similarities, with high values for SS, SK and $PP$ coinciding with low values for SSD and the selection of Ricker et al. (2016) using multiple criteria. The challenge is to set appropriate thresholds for each that allow effective discrimination.





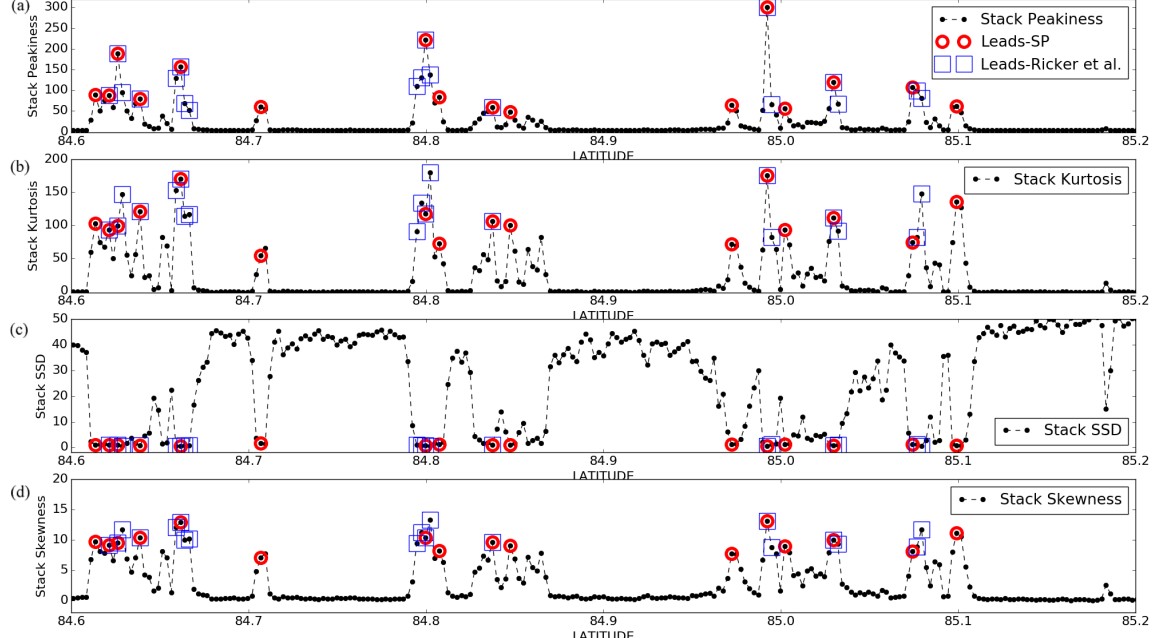

**Figure 7.** Illustration of general agreement between different indicators of leads. Pass is for CryoSat-2, and shows along-track profiles of a) Stack Peakiness, b) Stack Kurtosis, c) Stack Standard Deviation, d) Stack Skewness. Points selected according to a threshold on SP and according to scheme of Ricker et al. (2016) are highlighted.

### 2.2.3 Statistical techniques

There are now also some attempts at applying statistical techniques to the characteristics derived from the SAR waveform or RIP. The K-medoids approach of Müller et al. (2017) has been extended to work successfully with these data (Wynn et al.), and Shen et al. (2017) have implemented the machine-learning approach known as "random forest".

5     Challenges still remain in the recognition of melt ponds i.e. calm water patches lying on top of the ice during the melt season. Finally, it has to be stressed that, in defining absolute thresholds for SAR altimetry classifiers, developers must be aware that changes in the Level 0 to Level 1 data processing (e.g. application of Hamming weights in the Fourier analysis) can influence the performances of the method. Thus conclusions about the most appropriate waveform classification, and especially on the relevant thresholds are likely to be specific to the ground processing applied before the user sees the data.

10    ### 2.3 Validation of discrimination

Given the challenges of working in the Arctic, the narrow size of some of the leads and their tendency to change in time as well as to be advected by the currents, it is not surprising that usable ground truth data on the occurrence of leads are scarce. Indeed, the three principal sources of validation data for the waveform classification come from other remote-sensing techniques



### 2.3.1 Validation with optical sensors

There are a number of visible light and infra-red sensors that can provide sub-kilometre resolution imagery, with bright ice floes and dark leads (due to the much lower albedo of water). Many of the papers proposing waveform classification methodologies have simply overlain altimeter tracks on such optical imagery (Laxon et al., 2003; Guerreiro et al., 2017). Given currents of

$\sim$100m h$^{-1}$, such images must match the time of the altimeter overpass to within a few hours or else employ an ice drift model. Most of these studies have just provided an illustrative scene selected as a good example and have been manually edited for the effects of land and clouds, with no automation of the processing.

An interesting exception is the work by Poisson et al. (2018) on the validation of an Envisat classifier using fine resolution (300 m) output of the MERIS sensor. As both it and the RA-2 altimeter provide nadir measurements from the same satellite,

there is perfect spatial and temporal match-up. Variations in solar illumination angle meant that the water-leaving radiances derived from the MERIS scenes varied considerably; however the ratio between values at two different wavelengths provided a reasonably robust indicator. Their analysis applied to 42 selected scenes showed a good correspondence with the multiple criteria approach applied to the altimeter, but the selection of scenes was driven partly by the need to avoid clouds, which are spectrally quite similar to sea ice.

Lee et al. (2016) compared various SAR waveform classifications to 250 m resolution images from MODIS, with "visual interpretation" of the latter, rather than an automatic scheme. Although their implementation of "random forest" technique did not detect as many of the visible leads as the algorithm of Laxon et al. (2013), it created far fewer false detections. The most thorough investigation is by Wernecke and Kaleschke (2015), who compared altimeter waveform classification with images from MODIS, which occur every 1-2 days at high latitude. They analysed many months of data, with their "ground truth" being

based on a manual interpretation of the scenes, producing curves denoting the compromise between successfully detecting leads and increasing false lead detection as thresholds on classifiers are changed. They also evaluated the variance in sea level within a region according to which groups of waveforms were designated as "leads". For both these evaluations they found the best discriminator for the LRM waveforms to be maximal power. Note however, that their evaluation was carried out for the months January to March because those were the least affected by cloud, and so their assessment does not cover the period of ice melt.

However, it is to be expected that a myriad of usable data for development and classification of surface classification methods will be provided by Sentinel-3 satellites, since they carry both a delay-Doppler altimeter SRAL and an imaging optical instument OLCI. However, at the time of writing, no studies using Sentinel-3 on surface type classification have been published.

### 2.3.2 Validation with SAR

A different validation source is provided by high-resolution images from Synthetic Aperture Radar (SAR), which are slant-

viewing instruments recording surface backscatter intensity arising from suitably-orientated reflecting facets. This takes advantage of the different SAR scattering properties from various sea-ice surface conditions. Very flat and smooth areas, for example small open water areas, generate very specular backscatter characteristics; thus they appear very dark. In contrast, sea-ice affected areas exhibit a rougher surface and more diffuse reflections, which produce brighter pixel values. In general, the



brightness of the pixel is not only dependent on the surface conditions, but also on the transmitting frequency, the penetration depth and the incidence angle.

In contrast to multispectral visible and infra-red sensors, SARs are unaffected by clouds and illumination conditions, which enhance the opportunity to find spatio-temporally suitable images for a robust and reliable comparison process. As SAR

instruments cannot view the nadir locations observed by an altimeter, there is always the challenge of matching up SAR and altimeter observations from different satellites. The winds are the primary driver of sea-ice motion, with drift speeds of up to 40 km day$^{-1}$ (Hakkinen et al., 2008; Johannessen et al., 2013). Consequently it is necessary to match up observations to within an hour (e.g. (Passaro et al., 2017)) apply a model for sea-ice drift to correct the observations to the same time frame. Given the suite of different SAR instruments from different providers, and that power constraints limit the duty cycle (i.e. the proportion

of time for which the instrument operates), developing a large match-up database of SAR and altimeter measurements with appropriate sea-ice drift corrections is a large time-consuming task.

Müller et al. (2017) used short-wave C-Band SAR data from Sentinel-1A and Radarsat-2, and long-wave L-Band images from ALOS, with image co-ordinates shifted according to daily ice motion vectors from NSIDC (National Snow and Ice Data Center). Figure 8 illustrates the processing steps they used. First, a median filter is applied to the original image to

reduce speckle noise, and then a minimum filter is used to emphasise the dark areas (probable leads). The minimum filtering effectively expands the area ascribed as "leads", and is applied to overcome some of the uncertainties in the sea-ice motion. It uses a structure or kernel matrix that can have a variable size to accommodate different conditions. Then an adaptive threshold is applied to take into account illumination and contrast variations within the image.

The last step produces a linking of fragmented and adjacent open water areas. This is necessary because of local meteorolog-

ical and instrumental influences, (e.g. wind, refreezing or an insufficient pixel resolution) can cause small open water regions to brighten up and show a nosier scatter signature. As a consequence, leads and polynyas get divided after thresholding. To reconnect these areas, a mathematical morphological closing operation is used. It enlarges open water areas by mainly preserving their spatial extent and shape. Furthermore, it fills the gap between directly neighboring lead or polynya fragments. Similar to minimum filtering the closing operation is controlled by a kernel. In this case, the kernel size is set with regard to the pixel

resolution of the used images. For the comparison between the SAR images and the altimetry open water detection results, the binary converted SAR pixels are interpolated to the altimetry track coordinates using nearest neighbor interpolation. Further details on the method are published in Passaro et al. (2017).

Passaro et al. (2017) compared various characteristics of the SAR waveform stack (Figure 7) to a selection of SAR scenes. There is good general agreement between the various measures (SP, SSD, SS, SK and the multi-parameter test of Ricker et al.

(2014)); however, the SP-based classification was devised to identify a single point corresponding to each lead in order to avoid off-nadir returns, whereas SK and Ricker et al. (2014)) sometimes identify several points. This issue may be mitigated by suitable editing later in the processing (Sect. 3.4). It is difficult to set a threshold for SSD to work on its own, but it is one of the 6 parameters invoked by Ricker et al. (2014). A comparison with the dark areas in SAR images (Figure 8), within a tolerance of 400 m, shows the various methods to successfully detect ∼50% of leads found in SAR images, but also that around half of

those leads detected in the altimetry were not represented in the SAR images.

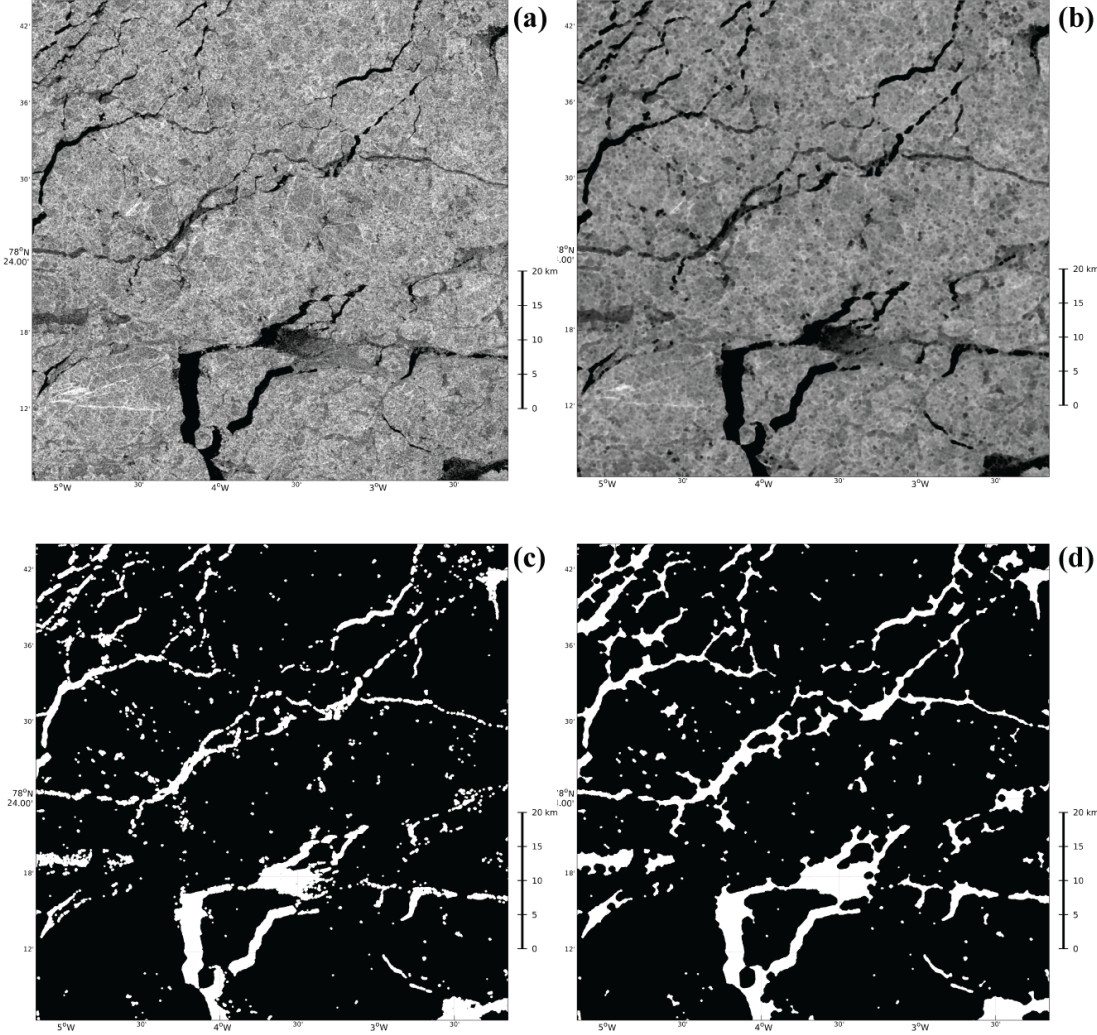

**Figure 8.** Subset of a Sentinel-1A SAR image: (a) the original grayscale SAR image after SAR preprocessing; (b) the same image after 5x5 pixel median and minimum filtering; (c) the binary image after adaptive thresholding; (d) final image after morphological closing operation giving open water in white and sea ice areas in black.

Changes in selected thresholds or characteristics used involve a compromise between successful detection and increasing numbers of false detections, e.g. in the analysis by Passaro et al. (2017), SP detected 9% more leads than the method of Ricker et al. (2014), but also yielded a similar proportion more of false detections. This could be a problem of the SAR image





resolution, since Sentinel-1 is not able to distinguish leads that are narrower than 40 m, which is still large enough to be the dominant return in altimetric waveforms (Passaro et al., 2017).

### 2.3.3 Validation through dedicated aircraft campaigns

A convenient way to acquire coincident altimeter and image data is to carry both instruments on the same aircraft platform.

Such data could be used for the validation of surface classification, since coincident measurements would eliminate the need to compensate for ice drift between acquisitions. To date, there have been two large airborne campaigns flying suitable instruments: NASA's Operation IceBridge (OIB) and ESA's CryoSat Validation Experiment (CryoVEx). However, the aim of the missions has been to provide freeboard, and snow and sea ice thickness estimates, so the data from imaging sensors have been used as an integral part of surface classification, rather than for validation.

Among the many airborne instruments often used during OIB are the Airborne Thematic Mapper (ATM, a conically scanning laser altimeter), a Ku-band radar altimeter and optical systems such as CAMBOT and Digital Mapping System (DMS) (Kurtz et al., 2013). Onana et al. (2013) developed a lead detection technique for data from the high-resolution DMS, which Kurtz et al. (2013) used in a comparison of lead/floe/ocean classification by the ATM. However, the latter instrument is a non-nadir pointing laser instrument, so the characteristics of its return signal are very different from a spaceborne nadir-pointing radar

altimeter; for example, very flat surfaces (such as leads or thin new ice) give little return to obliquely incident lidars. Thus visual imagery data provides the main source of reference data, with the analysis by Kurtz et al. (2013) using a surface classification scheme based on interpretation of data from the CAMBOT or DMS systems (Onana et al., 2013).

A similar approach is taken by King et al. in comparing CryoVEx radar and laser altimeter data with coincident aerial photography. They apply a lead detection algorithm to data from the airborne laser scanner (ALS) - in this case taking lowest

elevations to originate from leads. However, for ASIRAS (a CryoSat-like airborne SAR altimeter), they do not even try surface classification from its waveforms, but proceed straight to evaluation of freeboard. Aerial imagery is simply used to determine a possible bias between ALS and ASIRAS by manually selecting leads where the two should measure the same elevation. Previously, the ATM had been used to provide elevation data synoptic with an Envisat pass; Connor et al. (2009) showed visually the alignment of RA-2-detected leads with some changes in surface height from the ATM, but did not automate the

analysis to quantify how well the altimeter classification worked.

A rare direct comparison is shown in Fig. 9 indicating a good correspondence between high stack peakiness values (from a CryoSat-2 overflight) and the presence of large leads (as determined from aerial imagery). OIB has recently recorded some airborne data coincident with Sentinel-3A overflights; it is hoped that this will help the tuning of surface discrimination algorithms for that instrument.

## 30  3   Waveform retracking

The shape of a waveform is mainly governed by the reflections from multiple parts within the instrument footprint, which are individually at different ranges from the altimeter. The key measure of interest is the mean range of those facets at nadir,





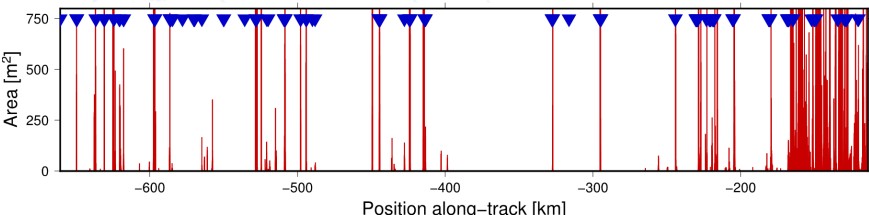

**Figure 9.** Comparison of OIB data with CryoSat-2 underflight in 2011. Areas of individual leads on the nadir track were estimated from digital photos (y-axis of display limits values shown to 750 $m^2$). Blue triangles show detection of leads by $PP$ calculated from waveforms, which match with 86% of leads over 250 $m^2$. [Image courtesy of Lars Stenseng, DTU.]

which is determined by fitting a model shape to the waveform data, noting that the observed signal will also contain elements of thermal noise (an additive component from the altimeter electronics) and fading (multiplicative) noise (Fu and Cazenave, 2001) . Over the open ocean, the waveform shape conforms to the Brown model (Brown, 1977) for which a number of algorithmic approaches exist e.g. (Amarouche et al., 2004). Alternative models are used for radar echoes from ice floes and from leads, and these are described below. In some cases the model has a mathematical form derived from the expected statistics of the physical processes producing the return; in other cases the "model" is empirical, simply based on getting a robust estimate of the delay in the return signal.

### 3.1 Waveform retracking for ice floes

Unbroken, but rough sea ice is a diffuse reflector, and so produces altimeter returns, for both LRM and SAR mode, that are similar to those observed over the ocean, although there may be differences due to volume-scattering within a snow layer on top (see Figure 3). Consequently, some approaches for determining range to ice floes have a strong inheritance of the physical retrackers used over the ocean, although others are more empirical.

### 3.1.1 LRM retracking for ice floes

In the early days of sea-ice altimetry, Scott et al. showed that the ICE-1 mode of ERS-1 yielded more reasonable measurements over non-ocean surfaces than were achieved by fitting the Brown model. This is because the echoes from many non-ocean surfaces often do not fit that theoretical model, whereas ICE-1 (Bamber, 1994), based on OCOG (offset centre-of-gravity) retracker, is much more robust as it only considers the position of the centre of the waveform. The same OCOG retracker was later implemented in the ERS-2 RA, Envisat RA-2 and CryoSat-2 ground processors, and has been used in many papers on sea ice thickness ((Laxon et al., 2003; Giles et al., 2007; Schwegmann et al., 2016).

An alternative is TFRMA (Threshold First Maximum Retracker Algorithm), which instead of considering the centre-of-gravity utilises the position of the maximum power of the waveform. However, the main characteristic of TFMRA (Helm et al., 2014) is its ability to select the first local maximum instead of some later peak, which could be due to off-nadir leads. Guerreiro et al. (2017), and Paul et al. (2018) both used TFRMA for retracking ice floe echoes from Envisat.



### 3.1.2 Empirical SAR waveform retracking for ice floes

The first published sea-ice thickness retrievals from CryoSat-2 relied on experimental retrackers for ice floes. However, the OCOG retrackers were soon superseded by more empirical threshold retrackers (such as TFMRA), whose main advantage is that they are simple and easy to realise, yet match well to independent validation data. Such an approach is efficient at

eliminating the effect of maximum peaks later in the waveform due to off-nadir leads, which is an issue for SAR waveforms too (Armitage and Davidson, 2014). TFMRA has since been used in several CryoSat-2 sea-ice studies, with varying thresholds for the retracking point (Hendricks et al., 2016; Ricker et al., 2017; Tilling et al., 2018).

Snow lying on the sea ice is a major concern, not simply because it is a delay correction to be applied (see Sect 4.2.2), but also because of its effect on the leading edge of the waveform. Ricker et al. (2014) showed that the choice of threshold

affects estimates of freeboard, but that this effect is spatially relatively constant. However they do suggest that varying the threshold according to surface conditions may be necessary in the future. Kwok raised the same concern based on airborne measurements and modelling. Due to the expected sensitivity of the leading edge to snow, Kwok and Cunningham (2015) used an OCOG-style retracker taking the centroid of the waveform as the retracking point. However, TFMRA remains the most used retracker today for SAR waveforms originating from ice floes.

### 15  3.1.3 Physical SAR waveform retracking for ice floes

In physical waveform retracking, the altimeter radar range is estimated by fitting the received return waveform with a model which best matches the received waveform shape and which is based on the physics of the electromagnetic interaction between the transmitted pulse and the scattering surface. Also this waveform model must incorporate, as much as possible, all the signal processing which has been applied on-board and on the ground.

For SAR altimetry processing, many waveform models are currently available in the literature which can be characterized to have either only a numerical solution or to have an analytical solution as well. Kurtz et al. (2014) developed an analytical form and showed how the SAR waveform shape and position (especially the 50% power level varied with angular backscattering efficiency and the S.D. of height variations. The SAMOSA SAR waveform model (Ray et al., 2015a, b) also belongs to the category of models with analytical solutions. It has been derived originally for open ocean thematic applications; it is widely

used over this type of surface and is the standard SAR waveform model for ocean retracking of Sentinel-3. Further, being an analytical model, it has the versatility to be very easily adaptable to any scattering surface once an appropriate scattering model of the surface has been incorporated in its formulation.

### 3.2 Waveform retracking for leads

Neither LRM nor SAR waveforms over leads correspond to their open ocean equivalents (see Figure 3), and due to differing

surface roughness statistics for open ocean and sea ice (Rivas et al., 2006), alternative retracking approaches are required.



### 3.2.1 LRM model for leads

The first attempt to retrack lead waveforms for range determination in LRM missions was based on Laxon (1994a), further implemented by Peacock and Laxon (2004) and included in the official ESA products from Envisat and ERS-2. The algorithm was based on a threshold retracker, while waveforms classified as open ocean were retracked with the standard Brown-Hyane

(BH) model (Brown, 1977; Hayne, 1980).

As a threshold retracker works by noting when power exceeds a certain fraction of the maximum (50% in the case of Peacock and Laxon (2004)), it requires linear interpolation between adjacent waveform samples within the leading edge (Gommenginger et al., 2011). The problem of using such a method with a fixed threshold is the assumption that the retracking point falls at the half-power point of the apparent leading edge, which in the case of peaky waveform is severely undersampled, as noted

already in the case of oceanic waveforms at very low sea state (Jenson, 1999).

Giles et al. (2007) also adopted a dedicated retracking strategy for lead waveforms, by dividing the modelled echo into a Gaussian, for the leading edge, and an exponentially decaying function to describe the tail. The strategy was originally developed for airborne radar altimeter waveforms. This model was also adopted by the Sea Ice CCI project Phase 1 for Envisat RA-2 (Ivanova et al., 2014), but was later replaced by a TFMRA scheme for both lead and sea ice surfaces in Phase 2 (Paul

et al., 2018).

### 3.2.2 SAR retracking for leads

Threshold algorithms such as TFMRA (Helm et al., 2014) have been used to retrack SAR altimeter returns across both leads and floes, accepting that there is a great difference in waveform shape, and thus there will be an offset between the measurements over the different surfaces. However, now there is increasing push to use physical models for deriving information over leads.

As the returns of a SAR altimeter over a lead are even more peaky than for LRM (see Figure 3), it is beneficial to apply zero-padding in the processing to acquire finer resolution within the waveform (see Figure 10). This is fully consistent with the process to estimate the range path delay within the instrument during the internal calibration (CAL1) or transponder calibration: in this case the signal is always highly oversampled in order to estimate the path delay within a few mm. Dinardo et al. (2017) have also suggested that the processing scheme could be configured to forego the multi-look capability of SAR altimetry and

use a single (nadir) look in these circumstances, as this will be sufficient to fit the peaky specular echo. Such an approach would have the benefit of a faster computational speed.

Further, it is important to point out that, when retracking the SAR altimetry data over sea-ice, different Level1b processing baselines can be implemented in order to identify that which is most appropriate for the procedure to retrieve sea-ice freeboard. Amongst these options are the application of zero-padding in the range dimension (leading to waveform oversampling by

factor of two), application of a weighting window on burst data in the azimuth direction prior to the FFT, and extension of the radar window by a factor of two. For CryoSat-2 baseline-C products, a Hamming window is used, but other authors (e.g. Smith (2018)) propose alternative weighting windows more tailored for applications over sea ice. The effect of these processing options on the final quality of the freeboard can be significant and needs to be properly assessed.



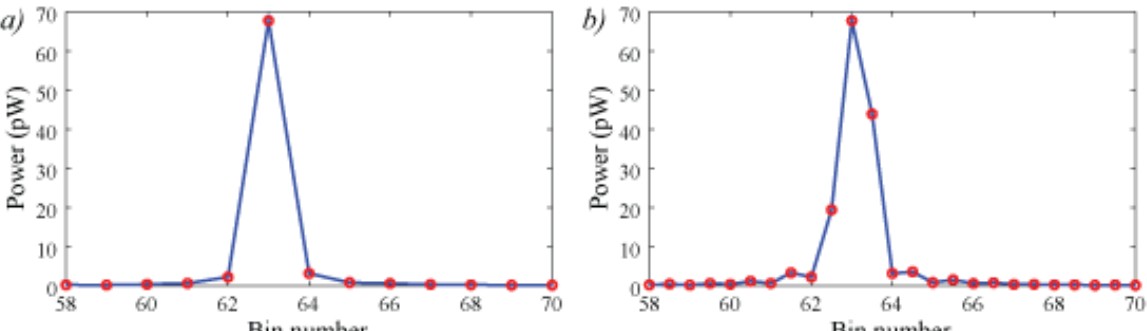

**Figure 10.** The effect of zero-padding upon the rendition of a CryoSat-2 waveform over a lead within the sea-ice. a) Without zero-padding, the specular waveform is heavily under-sampled (only one range sample within the main peak). b) With zero-padding (enabling the FFT to produce more frequent samples within the waveform shape), the peak is better represented, showing the asymmetry in the echo. This allows a more precise estimation of the timing associated with the 50% or other thresholds, reducing the jitter noise in the determination of the range. [In both cases, the full waveform corresponds to bins 1 to 128, with the panels being focussed in to show the details of the specular peak.]

### 3.3 Unified models for physical retracking

Empirical algorithms (such as OCOG and TFMRA) can be readily applied to all waveforms, but the very different shapes for radar echoes from leads and floes mean that the "retrack point" is effectively different for the two, and thus a relative bias needs to be determined (Giles et al., 2007; Laxon et al., 2013; Armitage and Ridout). Similarly, if different physical models are used for the two sets of waveforms (ice floes and leads) then a bias may exist between the two, which needs to be estimated and removed. To use a single physical retracker across these different surfaces requires that it be based on a shape model able to accommodate correctly both specular and diffuse waveforms.

Recently, two approaches have been proposed and applied to LRM missions. They are both based on a flexible approach to the BH model, in order to adapt the fitting process to include peaky echoes. They build on the heritage of Jackson et al. (1992), who showed that the surface roughness, expressed as mean square slope ($mss$) influences both the slope of the trailing edge and the location of the retrack point.

The key feature of Poisson et al. (2018) is the incorporation of $mss$ in the model of the flat surface response, resulting in a modified-BH functional form in which the $mss$ is an unknown to be estimated together with the usual BH parameters (see Fig. 11). Passaro et al. (2018) adapted the ALES retracker (Passaro et al., 2014) (which uses only a portion of the waveform around the leading edge) to adopt the value for the trailing edge slope coming from a prior estimate from the BH model. Both the Poisson et al. (2018) and Passaro et al. (2018) retrackers utilise an adaptive window at some stage in the estimation process: this feature focusses the fit on the leading edge, in order to avoid spurious contributions from the trailing edge. Poisson et al. (2018) found it challenging to demonstrate the continuity of their retracker across different surfaces, because those waveforms designated as "ocean", "lead" or "floe" tended to be well separated with many "unclassified" waveforms inbetween.





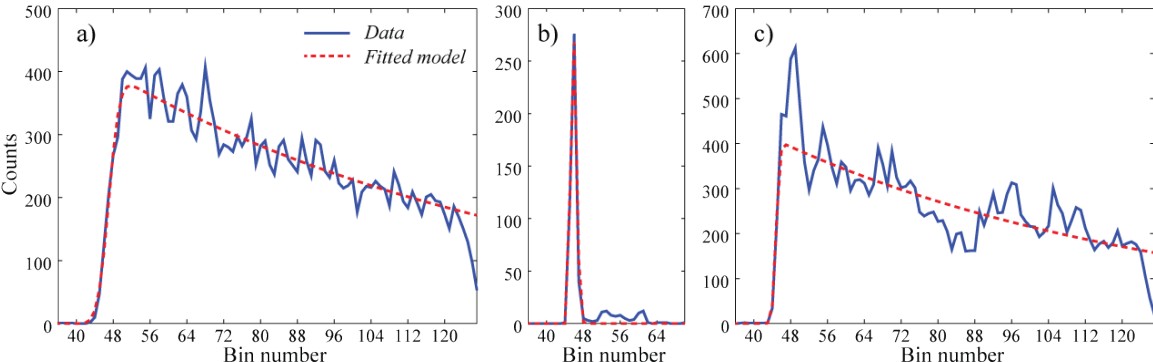

**Figure 11.** Illustration of the application of a flexible retracker model to accommodate waveforms from a) ocean, b) leads and c) floes. Data are from Envisat/RA2, with the waveform model described by Poisson et al. (2018) using $mss$ as an extra parameter principally affecting the trailing edge of the waveform. [The fitting process ignores the first four and last four waveform bins.]

Similarly, for SAR waveforms, Dinardo et al. (2017) has proposed the adaptation of the SAMOSA model to operate in coastal waters by involving $mss$ as an additional parameter, and also setting a more appropriate choice of the initialization that gives greater resilience to strong non-central returns, whether from land or off-nadir leads. This new retracker, referred to as SAMOSA+, can discriminate between return waveforms from diffusive and specular scattering surfaces, enabling appropriate retracking to be carried out. It has been applied successfully to retrieve sea-ice freeboard from CryoSat-2 SAR data.

### 3.4 Quality control: Further editing of data

The waveform classification procedure applies many aspects of quality control, in that data that do not conform well to the expected model for "ocean", "ice floe" or "leads" are rejected from the retracking process. This amounts to a point-wise editing of the data according to thresholds on $\sigma^0$ and $PP$ for example. There are further aspects to the editing that consider waveforms within the context of their neighbours in order to ascertain whether the returns belong unambiguously to one surface type or another, and are thus open to quantitative interpretation. Two related effects are discussed below, where the effects manifest themselves very differently: snagging is a response to bright targets away from nadir that generate peaks at a longer delay than nadir (i.e. within the trailing edge), whereas azimuth ambiguity is an effect particular to SAR processing that misconstrues a nadir return as though from a slant view and produces a peak before the leading edge.

### 3.4.1 Snagging effect within altimeter data

In the calculation of sea surface height, there is the assumption that the range recorded on-board the satellite is that to the nearest reflecting surface, which will generally be at nadir. However, the signal from a strongly reflecting lead will dominate the return signal for many consecutive waveforms (Figure 12). The retracking algorithms tends to follow such a feature leading to large errors in the estimates of surface height, with the distance from such a "bright target" tracing out a hyperbola in the



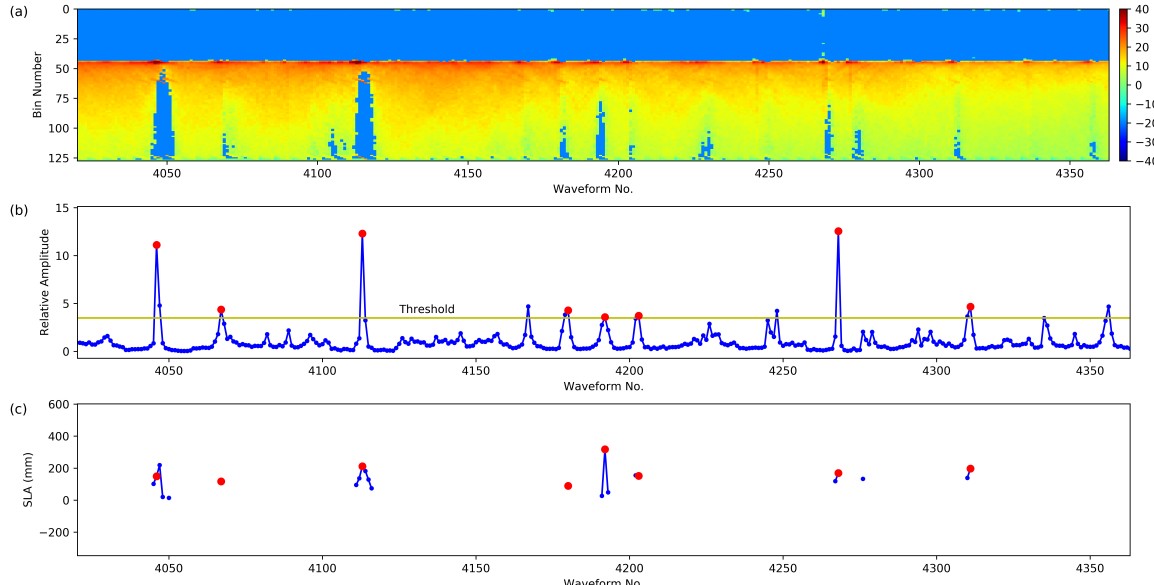

**Figure 12.** Editing ENVISAT RA-2 waveforms over Arctic leads and floes. a) Variation of waveforms along the altimeter track (with amplitude given in dB to emphasise the variation) b) Amplitude of peak intensity relative to a 21-point running average of that amplitude. c) Sea level anomaly (SLA), showing that there is greater consistency in the selected lead observations (red points) than achieved by keeping multiple records near the lead (blue lines from Giles et al. (2007) retracker).

waveform data (Gomez-Enri et al., 2010). This phenomenon is referred to as "snagging" by Peacock and Laxon (2004) and was given the name "off-nadir hooking" with application to radar altimetry over rivers (da Silva et al., 2010; Maillard et al., 2015). Using high-resolution MODIS imagery coincident with an Envisat track, Connor et al. (2009) have shown that reflections from a lead more than 1 km off the sub-satellite track can dominate the signals. Range errors related to returns from off-nadir leads

have been shown to also occur for SAR altimetry data from CryoSat-2, with an underestimation of the sea surface height by 1-4 cm, and strong biases in ice thickness estimation (Armitage and Davidson, 2014).

To reduce this effect and improve the accuracy of surface height estimation, Gomez-Enri et al. (2010) used a modelling approach, in which they replicated bright target features in the waveform sequence, and subtracted them from the waveforms to assist the retracking process. An automatic technique was proposed by Quartly (1998) to fit hyperbolic features within

waveform data. da Silva et al. (2010) investigated the snagging effect in ERS and Envisat data over rivers and lakes, by modelling and removing the response of off-nadir reflectors. This method has been applied to correct the altimeter measurements over narrow rivers, but could not describe the off-nadir distortions over large channels or lakes (da Silva et al., 2010). The correction method also required visual inspection of range measurements close to river banks and open water, and could not be implemented automatically. Maillard et al. (2015) applied a pattern recognition technique to fit the sequence of surface height

measurements over rivers. However, its has limited application to the measurements in sea ice, as the location and shape of leads are not known a priori.





Poisson et al. (2018) improved their sea level estimates by a data-editing approach that consisted of detecting the waveforms with strong reflection in the nadir direction, and then discarding the neighbouring waveforms. This editing approach improved the retracker accuracy by discarding potentially biased range measurements around strong nadir reflection points, and is illustrated in Fig. 12 for RA-2 Ku-band waveforms over Arctic leads and floes. Strong nadir reflections produced sharp spikes,

which were automatically discriminated from the rest of waveforms (Fig. 12b). This procedure discarded sea level measurements around such spikes (blue dots in Fig. 12c) as they were likely to produce biased results. In contrast, the output of Giles et al. (2007) retracker in Fig. 12c was affected by "snagging" effect with larger range estimates produced in the neighborhood of spikes.

### 3.4.2    Azimuth ambiguity effect within SAR data

The off-nadir ranging effect, as described for the LRM mode, has less impact on the SAR echoes. This is due to the along-track beam-limited resolution, which also reduces the backscatter from across-track points. But it nevertheless remains present and still needs to be filtered out. However, a much more important side effect must be considered while operating in SAR mode over sea-ice, which is known as the "side-lobe effect" or "azimuth ambiguity" effect. This effect of the SAR processing occurs while measuring low backscattering surfaces with nearby high backscattering surfaces along the track of the satellite, which is a

situation typically encountered while measuring floes surrounded by leads. In this case, the weighting provided by the synthetic beam's antenna pattern cannot fully compensate for the highly contrasting backscatter strengths. In practice, this phenomenon introduces spurious power before the leading edge of the nadir backscatter (see Fig. 13); whereas the snagging effect (in the previous sub-section) introduces spurious backscatter after the leading edge, with the tails of its hyperbolae tending towards longer range. These early "ghost" peaks along the waveforms can confuse the retrackers, or even corrupt the nadir peak.

Several strategies can be used to overcome this phenomenon, the first of which is the application of Hamming filtering before the Doppler beam processing. This is the case for the ESA CryoSat-2 products, but not for the Sentinel-3 products from ESA and EUMETSAT. This filter is not systematically applied on all the products because it has some minor effects on the final waveforms, including a slight reduction in the along-track resolution (Scagliola et al., 2014). Two alternative approaches can be considered for products which have not filtered out the side-lobes: the first one consists simply in eliminating the waveforms

that contains multiple peaks and the second one aims to localise the peak that corresponds to the nadir and which is the one to be retracked. Only the waveforms that show clearly distinct peaks can be kept, but they may provide very useful intermittent measurements in highly fractured sea-ice areas.

### 3.4.3    Ensuring Consistency in Space and Time

When compiling a long-term dataset spanning multiple instruments and various retrackers, it is essential to minimise the biases

between constituent parts. Although a mean bias between instruments may be determined on a global basis or via dedicated calibration sites, it is important that these offsets are also evaluated in the Arctic context. This is critical because some studies have used different retrackers for waveforms from floes, leads and open ocean, whilst others have adopted one retracker, such



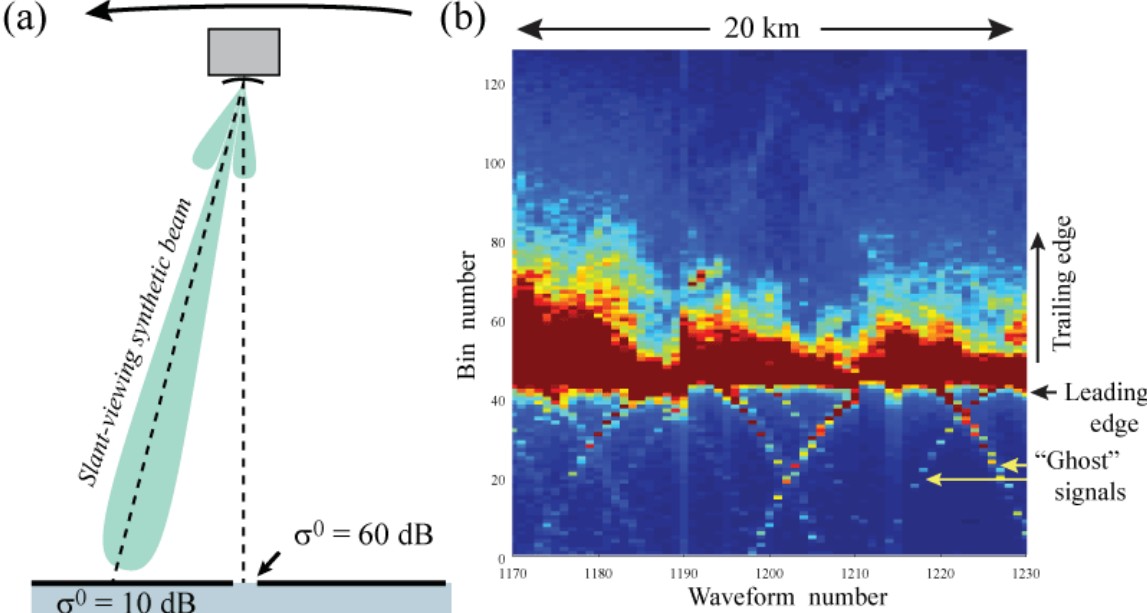

**Figure 13.** a) Schematic showing that when a synthetic Doppler beam is looking at an off-nadir ground cell in a slant view, the finite size digital filter effectively has weak sidelobes directed elsewhere. If there is a much brighter source near nadir (symbolized by the gap in the sea-ice cover), its contribution will also be recorded. As this strong return is nearer than that from the intended slant view, upon SAR Stack range compensation for the expected geometry, it will appear as parabolic arch ahead of the waveform leading edge. b) Illustration from Sentinel-3A SAR waveforms over the Arctic of the resultant "ghost" images ahead of the leading edge.

as TFMRA, but noted that the very different waveform shapes essentially mean that they have different retrack points (Giles et al., 2007).

    Ideally the utilisation of a unified model (see Section 3.3) should avoid any artificial change in sea level associated with the ice edge (as the sea level is derived from open ocean waveforms on one side and predominantly from leads on the other).

5    Further challenges exist in the context of multi-mission datasets spanning several decades, as there have been changes in technology (LRM and SAR) and differences in processing methodology between missions (e.g. use or not of Hamming weighting). Of particular note is that AltiKa operates at a different radar frequency to all other missions (Table 1); thus volume scattering from snow is more significant at this frequency leading to changes in the retrack point over ice floes (Guerreiro et al., 2016).

10    A critical issue in the area of consistency is the quality and reliability of the geophysical corrections (see Section 4), since some missions enable more reliable corrections than others e.g. due to the presence of a second radar frequency or having an on-board microwave radiometer. Finally, we note that the changes in sea-ice cover on both seasonal and interannual scales necessitate extra care when analysing for long-term changes in sea level. The time series in Fig. 14 are made from data downloaded from the Radar Altimetry Database System (RADS) using default editing (for retrieval of sea level from the open





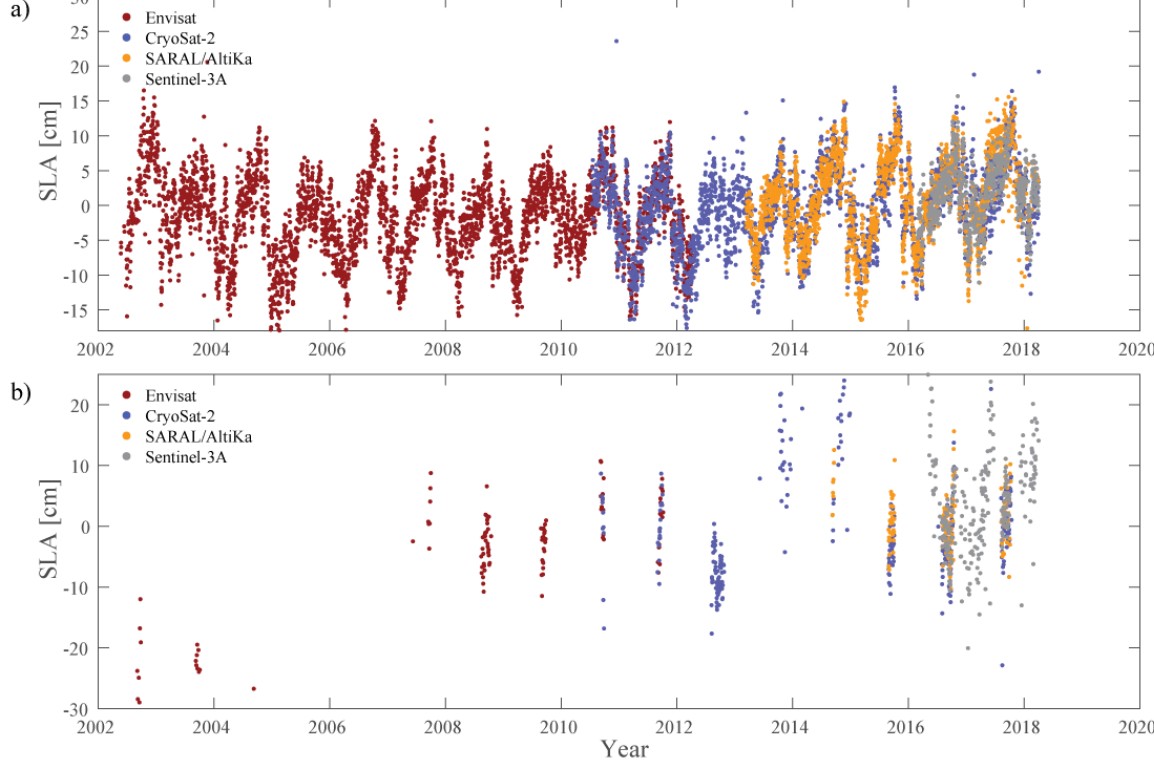

**Figure 14.** Time series of Sea Level Anomaly (SLA) from 4 different altimeters for a) Greenland Sea, b) Beaufort Sea. [Image courtesy of Heidi Ranndal, DTU.

ocean, not from within leads) and wet tropospheric corrections from the ECMWF. Data were downloaded for four different missions in two areas: the Greenland Sea [10°W-10°E,75-80°N], and the Beaufort Sea [150-130°W,75-80°N]. The seasonal availability of data is very different for the two regions.

Whilst the Greenland Sea is rarely ice-covered, the Beaufort Sea is ice-covered to some degree throughout most of the year. This causes a distribution of sea level anomaly (SLA) estimates for the latter region that is very biased towards the summer months (see Fig. 14b), since data from winter and spring are flagged and discarded. In fact, most of the available data in the Beaufort Sea are obtained in late summer and early autumn, where the sea ice has reached a minimum, and where there is a high risk of inaccurate range estimates due to the presence of melt ponds on the sea ice. It might be possible to recover more of the data by lowering the requirements on their quality and with different processing. Nevertheless, this seasonal variation in data quantity may affect estimation of annual and inter-annual signals. In addition, the amplitude of the seasonal variation seems to be decreasing slightly in recent missions compared with the Envisat period. This could be due to the higher resolution of the altimeters or because of decreasing sea-ice cover.





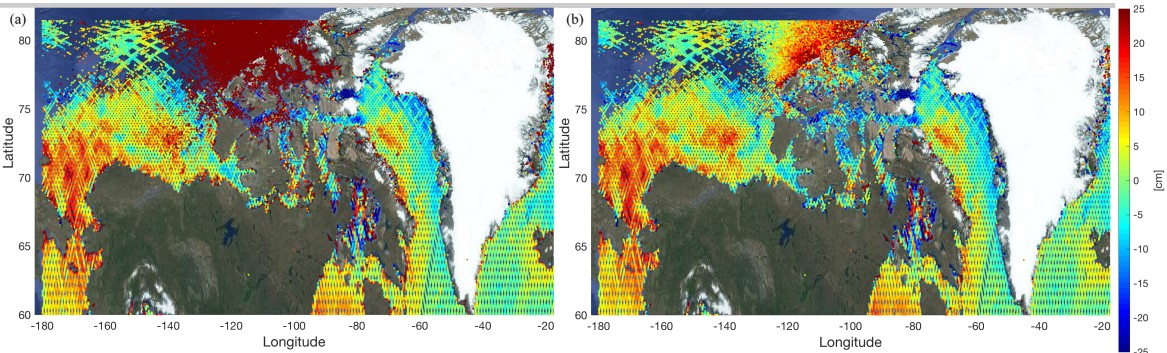

**Figure 15.** Mean sea level anomaly for Sentinel-3A data spanning cycles 8-23 (15 months), calculated using the WTC from a) the on-board MWR, b) the ECMWF model.

## 4 Corrections and references needed for inferring precise geophysical values

### 4.1 Determining sea level

For measuring sea level, and investigating temporal changes or spatial variations associated with geostrophic currents, it is essential that no bias remains between different altimeters or between any altimeter's processing over open ocean over leads.

Not only must the measurements be consistent, but a host of corrections need to be estimated and applied rigourously. The particular constraints within the Arctic environment are discussed below.

### 4.1.1 Atmospheric corrections

In standard open ocean altimetry processing (Fu and Cazenave, 2001; Quartly et al., 2017) there are three atmospheric components that alter the propagation speed of the radio waves and one atmospheric process that affects sea level. The three

corrections are the ionosphere correction, the dry tropospheric correction (DTC) and the wet tropospheric correction (WTC). The ionospheric correction is to compensate for the effect of free electrons in the propagation path, which is a very minor effect at polar latitudes, and thus output from a model may be reliably used. The DTC relates to the mass of air, and thus the surface pressure, with high-resolution meteorological reanalyses being used to calculate this correction.

Over open ocean, WTC is normally calculated from the measurements by the on-board microwave radiometer (MWR);

however the brightness temperatures recorded by the MWR do not provide reliable estimates of atmospheric moisture over ice, and thus output from numerical reanalyses is again preferred (see Fig.15). Indeed, CryoSat-2, which initially had a purely cryospheric focus, does not carry an MWR. The Dynamic Atmospheric Correction (DAC) is a modelling of the sea surface response to the preceding time series of pressure and wind, and is also calculated from a meteorological model. The full DAC is normally implemented over the ocean, but for the sea level under ice floes only the simple "static" response to the pressure

field is applied.



**Table 2.** Alias period (in days) for recent and near-future satellites

| Tidal constituent | Envisat, AltiKa (35-day repeat) | Jason-3, Jason-CS/Sentinel-6 (9.9156-day) | CryoSat-2 (369-day repeat) | Sentinel-3A, -3B (27-day repeat) | IceSAT-2 (91-day repeat) |
|---|---|---|---|---|---|
| M2 | 94.4 | 62.1 | 112.1 | 157.5 | 55.79 |
| S2 | Infinite | 58.7 | Infinite | Infinite | Infinite |
| K1 | 365.25 | 173.1 | 98.1 | 365.25 | 365.25 |
| O1 | 75.1 | 45.1 | 77.1 | 277.0 | 220.1 |

Note, the Jason satellites do not sample the Arctic because their orbital inclination is only $66°$, but are included here for completeness.

### 4.1.2 Tides and Mean Sea Surface

Arctic Ocean tides are generally difficult to determine as the data available are predominantly from sun-synchronous satellites. In particular, the S2 constituent, which can have a magnitude of 50 cm in the Arctic, is frozen in the orbit (i.e. the satellite always observes exactly the same phase of that tidal component) so it cannot be determined. Nor can sun-synchronous satellites

help us disentangle the aliasing of the K1 tide (see Table 2) and the annual sea level variation. Consequently the development of new and better tide models using hydrodynamic modelling is essential for the Arctic Ocean.

The first empirical tide model of the Arctic Ocean was derived from ERS-1 data by Andersen. Since then numerous ocean tide models have been derived. Stammer et al. performed a comparison for 8 state-of-the-art models with tidal constituents derived from some of the 240 Arctic tide gauges maintained by Kowalik and Proshutinsky (available via http://www.ims.uaf.edu/tide/).

The agreement found is still far from as good as that noted in comparisons with tide gauges at mid to low latitudes.

To calculate the freeboard, the surface height of the ice floe has to be compared with the expectation of where the sea level would be. This is inferred by interpolating the occasional measurements within leads of the sea level relative to the mean sea surface (MSS). The MSS is thus the largest correction, as it ranges over more than 40 metres across the Arctic Ocean, but an adequate model of the tides (including S2) is essential in order to provide an accurate MSS. Sentinel-3 adds crucial

information about the MSS north of Canada. However, the WTC correction based on MWR data must be avoided to prevent highly unrealistic values north of the Canadian Arctic Archipelago (see Fig. 15).

Several MSS products exist based on different satellite altimetry input data over varying periods. Differences and properties of these products are assessed in Skourup et al.. The state-of-the-art MSS fields covering the whole Arctic Ocean are the UCL13 and DTU15 MSS. (The coverage of the CNES/CLS15 MSS is incomplete, and it is therefore not generally applicable

for pan-Arctic studies.) The UCL13 MSS is provided to the user within the ESA CryoSat-2 baseline-C data products, and has been specially tuned to improve sea-ice freeboard retrieval. South of $50°$N the CLS 2011 global MSS is used, and the two merged between $50°$ and $60°$N. The DTU MSS surface is referenced to the 1993-2012 period and in the Arctic is derived from ERS-1, ERS-2, Envisat and almost six years (2010-2015) of CryoSat-2 baseline-B data. North of $88°$N the MSS is tapered



towards EGM2008 GGM (Andersen et al., 2016), which is a representation of the geoid.

## 4.2 Determining freeboard and ice thickness

### 4.2.1 Interpolating sea level anomaly and calculating freeboard

Surface elevations are determined for ice floes and water in leads relative to the ellipsoid describing the approximate shape of the Earth, with the appropriate range corrections, including MSS, being subsequently applied (see Section 4.1). Next, lead elevations are interpolated to retrieve the sea level anomaly (difference between the instantaneous sea surface height and the MSS) along the whole altimeter track. This sea level anomaly is subtracted from the sea-ice elevations to give the freeboard. Since freeboard is a relative quantity, geophysical corrections should not affect the retrieval; however, due to spatial variations in

the sea surface height between leads, interpolation errors are likely to occur. Therefore, applying geophysical range corrections (atmospheric, geophysical, MSS) is essential for minimizing such interpolation errors, which can lead to significant biases in the freeboard retrieval. The importance and impact of these range corrections on sea level anomaly and sea ice freeboard retrievals are discussed in Skourup et al. for the MSS and in Ricker et al. (2016) for geophysical and atmospheric corrections.

### 4.2.2 Freeboard to thickness conversions

The estimation of sea-ice thickness, $t_i$, from the altimeter-derived sea-ice freeboard is based on Archimedes Principle, assuming that the sea ice is freely floating (see Fig. 16). The application of this is common to all studies using altimeter data over sea ice in the scientific literature. Variations to a certain degree do exist in the parameterization of the conversion, which requires the densities of sea ice, $\rho_i$, sea water, $\rho_w$ and the snow load given by snow depth, $t_s$ and density $\rho_s$ as input parameters:

$$t_i = \frac{t_s.\rho_s + f.\rho_w}{\rho_w - \rho_i} \qquad (2)$$

where $f$ is the freeboard determined from the altimetry. However, aside from the uncertainty in the freeboard estimate, there is significant uncertainty in the values of the constants to be used. Whilst the density of the sea water is a a well-known parameter with little variation between 1023 to 1024 $kgm^{-3}$ (Wadhams et al., 1992), there is much less clarity about ice density, and snow depth and density. An ice-type dependent parametrization is common for sea ice density, which varies with the amount of brine or air bubble entrapped in the ice layer. The main difference is the higher amount of air bubbles in multi-year ice (MYI),

making its density lower than that of first-year ice (FYI). Alexandrov et al. (2010) report bulk densities for level-ice of 882±23 $kgm^{-3}$ (MYI) and 916±35.7 $kgm^{-3}$ (FYI), values that are commonly used in the freeboard to thickness conversion. However, it must be noted that these values are associated with level sea ice only: the mean thickness of sea ice also contains a significant fraction of deformed ice where the density is likely to deviate. Other solutions, e.g. explored by Kwok and Cunningham (2015) include single bulk ice density or thickness-dependent densities.





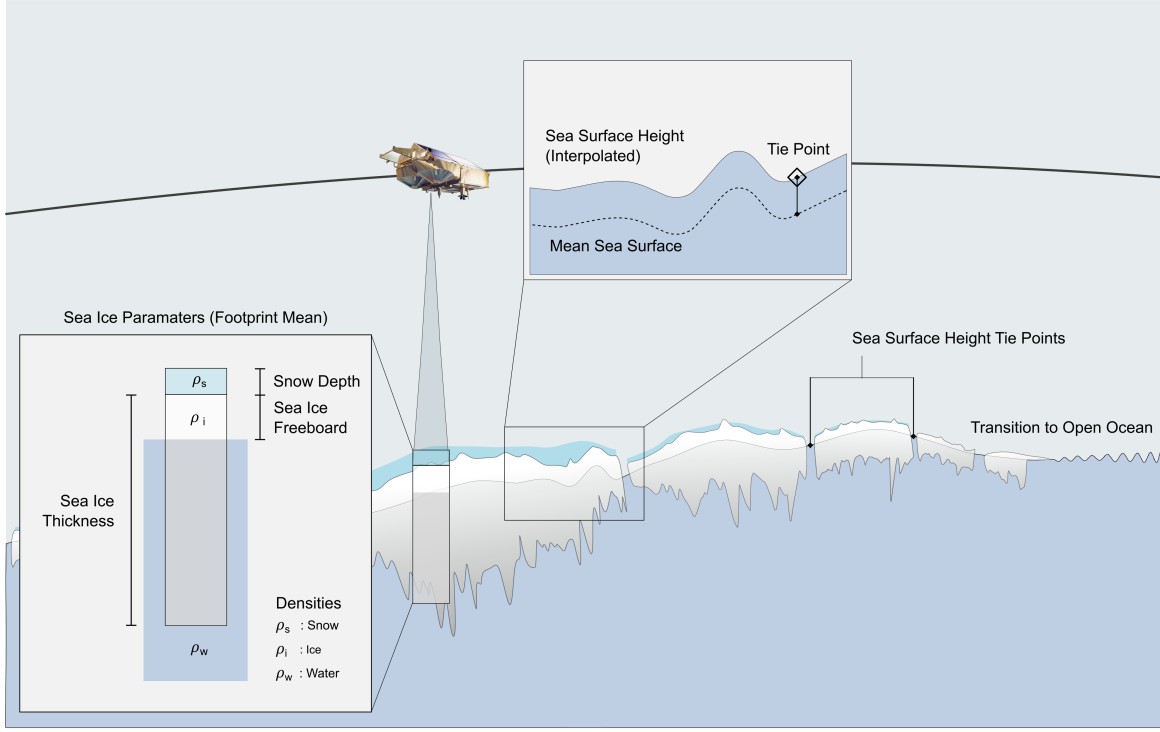

**Figure 16.** Schematic showing that an altimeter measures distance to ice surface (or the leads within it), so sea-ice thickness calculation must allow for weight of snow layer and the difference in densities of water and ice.

Snow depth is not a routinely observed parameter on basin scales, a limitation that introduces a significant error component in the freeboard to thickness conversion (Giles et al., 2007; Kern et al., 2015). The most common approach in calculating recent sea-ice thickness products (Laxon et al., 2013; Ricker et al., 2014; Kurtz et al., 2014; Kwok and Cunningham, 2015; Guerreiro et al., 2017; Tilling et al., 2018) is to replace the limited observational data with a snow load climatology compiled from in

5  situ observations in the period of 1954–91 (Warren et al., 1999). This provides monthly fields of snow depth and density in the form of a fitted two-dimensional quadratic function and a measure of interannual variability for both depth and density. Recent airborne observations have however shown that the climatology overestimates snow depth in regions of FYI by as much as 50% (Kurtz and Farrell, 2011; Webster et al., 2014). All snow climatology-based sea-ice thickness studies therefore apply a 50% snow depth reduction for areas of FYI.

10  While the ice-type based modification of climatological snow depth may compensate for a potential trend in snow depth on sea ice, the interannual and regional variability in snow accumulation will not be mapped correctly. There have been recent efforts to infer snow depth from the difference in observations by Ku- and Ka-band altimeters (Guerreiro et al., 2017), but these cannot furnish values prior to the launch of AltiKa in 2013. However Lawrence et al. (2018) claim some success using a combination of Ku-band radar (Envisat) and laser (ICESat); although the operation of ICESAT was intermittent (operating





for only a few months in the year), this could still contribute to a better snow-depth climatology for the 21st century. Field observations over large areas that study the relationship of freeboard and draft (Doble et al., 2011) are still required to reduce uncertainties in the freeboard to thickness conversion. In the future, snow depth from reanalysis products may be provided as an auxiliary data source to improve ice thickness retrieval from radar altimetry.

## 5    Comparison with in situ and airborne measurements

In section 3, we showed the use of in situ, airborne and other satellite data for validation of the waveform classification; in this section, we assess the quality of the overall physical retrievals as the results of classification, retracking and application of corrections. The first subsection deals with the oceanographic quantities, and the second with those relating to the sea-ice.

### 5.1    Sea level and currents

One of the first oceanographic applications of radar altimetry over the Arctic Ocean was the derivation of the marine gravity field for the permanently ice-covered regions. Laxon and McAdoo (1994) analysed the mean topography of the ocean derived from the elevation measurements in leads during a 35-day cycle of ERS-1. They showed that this mean sea surface conforms to the geoid and variations of the Earth's gravity due to density variations in the mantle (low spatial frequencies) and to sea floor topography (high frequencies). Such satellite-derived gravity fields were validated against airborne gravity surveys (i.e.

Canadian Geophysical Survey) and shown to conform very well in the Beaufort Sea.

Greater precision is gained from improved altimeter measurements, and better resolution acquired through the denser pattern of tracks in long-repeat orbits. ERS-1 had a so-called "geodetic phase" providing finer longitudinal sampling than in its usual 35-day repeat, but there has been a marked improvement in the Arctic marine gravity field modelling since the launch of the CryoSat-2 mission in 2010. With its 369-day repeat, it provides one cycle of geodetic mission data with 8 km global resolution

each year. The higher precision of these new sea surface height observations compared with observations from ERS-1 and Geosat means that these latter data are no longer used, which has resulted in a dramatic improvement of the shorter wavelength of the gravity field (12-20 km) and better comparison with marine gravity data (Andersen et al., 2015). The pan-Arctic altimetric gravity field DTU15 now surpasses the 2008 Arctic Gravity Field project compilation of marine gravy data from multiple sources, as can be seen from comparison with an independent gravity field from GOCE (Andersen et al., 2017).

The sea surface height signal can be decomposed into two components: eustatic (change of mass of the column of water) and steric (change of ocean density). Armitage et al. (2016) compared two estimates of the steric signal − integration of density profiles from Ice-tethered profilers (ITPs, (Peralta-Ferriz et al., 2014)) and altimetry minus gravity field from GRACE − and found a good correlation (R∼0.86). They concluded that the Arctic SLA variability is dominated by the seasonal cycle, with the first Principal Component capturing 38.7% of the total SLA variance.

A mean dynamic topography (MDT), that is, the height signature consistent with the mean surface geostrophic currents, has traditionally been derived using temporal averaging of a hydrodynamic models (such as TOPAZ or SODA). With developments in the accuracy of the MSS and geoid, the MDT can also be derived from the difference between them. The resultant MDT





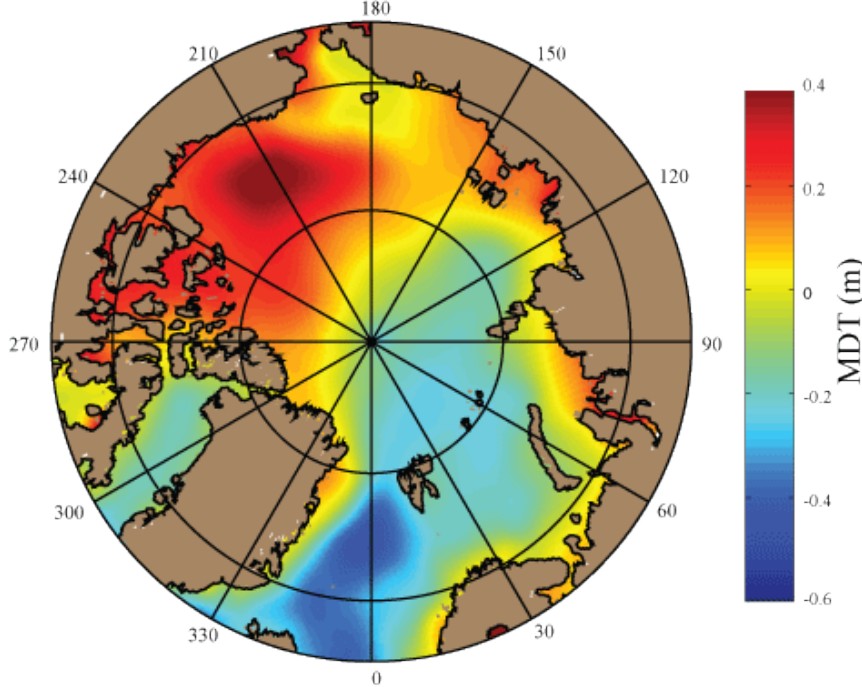

**Figure 17.** The DTU15MDT derived from the DTU15MSS and the EIGEN6-C4 geoid model

will consequently represent the temporal averaging period over which the corresponding MSS was derived. Farrell et al. (2012)
derived one of the first satellite-based MDT using ICESat; recently Andersen et al. (2015) derived one called DTU15MDT (see
Fig. 17) taking into account GOCE and CryoSat-2 data. Its main features are consistent with an MDT derived from the TOPAZ
model (Johannessen et al., 2014), including a signal larger than 0.3 m in the Beaufort Sea associated with the anticyclonic

Beaufort Gyre and a large-scale slope (∼0.6 m/1300 km) in topography from the Amerasian Basin to the Eurasian Basin
associated with the transpolar current (see Fig. 1a). Qualitatively these results agree well with others derived from satellite
altimetry (Kwok and Morison, 2016; Giles et al., 2012) as well as from ocean models (Koldunov et al.).

The first empirical tide model of the Arctic Ocean derived from the ERS-1 and ERS-2 altimeters (Peacock, 1999) was
validated against tide gauge measurements. Similar validations and comparisons have since been generalised to more recent

models that assimilate all low inclination radar altimeters (Cancet et al., 2018). Validation of an interpolated gridded tide
model against tide gauges is typically performed at the location of coastal and pelagic stations, with there being hundreds of
the former, but only tens of the latter. In general agreement between tide models or sea surface heights derived from radar
altimeters and tide gauge measurements is very good, particularly for locations away from bays and very shallow waters where
strong tidal signatures are present. Armitage et al. (2016) compared all Arctic tide gauge records (Holgate et al., 2013) with

15 more than 72 months of data for the 2003-2014 period covering measurements from Envisat and CryoSat-2 and confirmed this
good agreement for tide gauges in the Canadian Arctic, Barents Sea and Svalbard (R∼0.8-0.9), while correlations for stations





in the Kara, Laptev and East Siberian Seas were found to be lower (R∼0.5-0.7) due to the larger impact of seasonal runoff and to the proximity to river estuaries. Recently, Armitage et al. (2017) produced a 12-year time series of geostrophic currents in the Arctic and performed the first direct evaluation against in situ measurement by three acoustic Doppler current profilers (ADCPs) in the Beaufort Sea, showing significant correlation in speed for two out of three moorings and significant correlation

in bearing for one of them. Some of the differences are explained by the difference in footprint and timescale over which the data are collected for both techniques.

## 5.2    Freeboard and sea-ice thickness

There are several different methods to evaluate the satellite-derived sea-ice freeboard and thickness. Through the years of satellite altimetry, in situ observations, airborne campaigns, submarines, and drifting and moored buoys have all been invaluable

in measuring the sea-ice and snow properties. However, as it is difficult and expensive to operate in the harsh environment with cold temperatures and darkness during the Arctic winter, such observations are still sparsely distributed in space and time (see Fig. 18).

The various evaluation data sets have their own pros and cons with respect to spatial and temporal resolution. Airborne and submarine surveys cover larger areas, and usually represent short temporal scales (days or months). Whereas the moored and

drifting buoys represent point measurements over longer time scales, and provide information about seasonal variations.

### 5.2.1    Freeboard

There are two reasonably extensive datasets for direct satellite sea-ice freeboard evaluation. One is the freeboard values from NASA's Operation IceBridge (OIB), provided either as sea-ice + snow freeboard or sea-ice freeboard only. The other is from the airborne campaigns carried out as part of ESA's CryoSat Validation Experiment (CryoVEx), where ASIRAS (an airborne

version of the CryoSat-2 SIRAL altimeter) provides coincident Ku-band radar freeboard data. More recent campaigns (2016 and 2017) have included an airborne Ka-band radar altimeter (KAREN) to evaluate SARAL/AltiKa, and also to exploit the potential of a dual-frequency concept for future satellite missions. The processing of the KAREN data is still ongoing, so results can not be included here.

The airborne OIB and CryoVEx data are primarily used to evaluate the satellite radar freeboard, to investigate penetration

depths of both Ku- and Ka-band radars and the potential for snow-depth retrieval (Armitage and Ridout; Maheshwari et al., 2015; Guerreiro et al., 2016; Connor et al., 2009; Lawrence et al., 2018), and to examine the sensitivity to the choice of using different re-trackers (Kurtz et al., 2014; Ricker et al., 2014). These studies consistently conclude that the predominant signal from Ku-band satellite radars such as CryoSat-2 and Envisat corresponds to reflections from close to the snow-ice boundary. Other studies of CryoSat-2 (Ricker et al.; King et al.) used Ice Mass Balance (IMB) buoys and a combination of airborne

and in situ measurements of sea-ice and snow properties to show that the influence of snow cover on Ku-band penetration is not negligible in specific regions and/or snow conditions. Ricker et al. show that an unexpected increase in CryoSat-2 sea-ice freeboard was correlated with an exceptionally high snow accumulation in the beginning of the 2013 growth season at the north of Canada. In addition, King et al. found that the CryoSat-2 sea-ice freeboard and thickness in the Norwegian Arctic





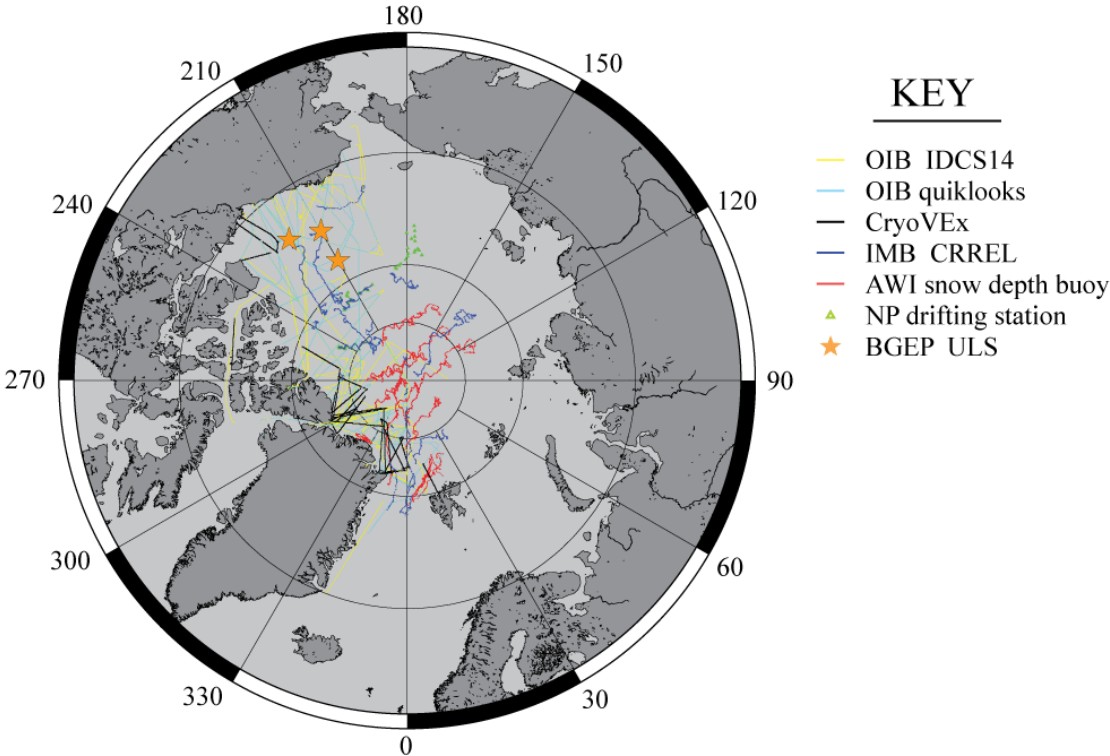

**Figure 18.** Map showing the limited in situ data available, with many sectors of the Arctic poorly sampled. OIB signifies flights as part of Operation IceBridge, with the IDCS14 product covering 2009-2012, and the quiklooks for 2013-2016.

spring (represented by thin sea ice, thick snow and widespread negative freeboard) were better represented by assuming no penetration of the radar even in cold conditions (<15°C).

A wider range of conclusions was found for the Ka-band radar altimeter (SARAL/AltiKa). Guerreiro et al. (2016) concluded that AltiKa's signals corresponded to reflection at the air-snow surface, Armitage and Ridout that they were almost half-way
5   between the air-snow and snow-ice surface, and Maheshwari et al. (2015) at the snow-ice surface. Lawrence et al. (2018) found a surface-dependent bias of the AltiKa radar freeboard against airborne snow freeboard. The use of different retrackers as well as the consideration of the impact of sea-ice roughness could explain some of these inconsistencies.

### 5.2.2 Sea-ice thickness

In the literature the most commonly used observations to evaluate satellite-derived sea-ice thickness are derived sea-ice thick-
10   ness from NASA operation IceBridge (OIB), sea ice + snow thickness by airborne electromagnetic (AEM) sensors and sea-ice draft. The latter is measured by upward-looking sonars (ULS) either from submarine cruises or moored buoys (for example during Beaufort Gyre Experiment; BGEP). A more detailed description of these data sets is found in Lindsay and Schweiger (2015) and Kwok and Cunningham (2015). As none of the above-mentioned observations measures the sea-ice thickness





directly, the satellite-derived sea-ice thickness needs to be changed into either draft, or sea-ice + snow thickness before comparison, with a priori assumptions of snow depth and/or densities of snow and sea ice. In addition, the OIB values are estimated from the sea-ice freeboard (sea-ice + snow freeboard minus the snow depth) according to Eq. 2. Thus, in order to compare the sea-ice thickness from satellite altimetry with in situ observations it is important to have consistent assumptions about snow
depth and the densities of sea ice, snow and water.

Despite these challenges and the many assumptions in the processing chain, recent studies (e.g. (Laxon et al., 2013; Kwok and Cunningham, 2015; Tilling et al., 2018)) find relatively good correlations (0.5-0.9) with mean differences -0.21 to 0.12 m between the CryoSat-2 derived sea-ice thickness products and the evaluation data in the central Arctic winter (October-March). Prior studies comparing ERS-1 and ERS-2 with draft from submarine cruises (Laxon et al., 2003) found an almost one-to-one
agreement. These results are within the expected uncertainties of measurements and no systematic biases were found. Often the correlation between OIB and CryoSat-2 sea-ice thicknesses are found to be lower than with submarine, moored buoys and AEM observations. The cause of this is still unclear and subject to further investigation (Laxon et al., 2013; Kwok and Cunningham, 2015).

### 5.2.3 Validation of algorithms

Last but not least, total thickness of sea-ice + snow by AEM sensors and sea-ice draft from moorings have also been used to evaluate and improve various steps in the processing chain, such as the Envisat freeboard retrieval by including information about surface roughness (Guerreiro et al., 2017). Submarine, airborne, and buoy data have also been used in sensitivity studies to investigate the parameters (snow depth, densities of sea ice) used in the freeboard to sea-ice thickness conversion (see Eq. 2). No consistent results were found, as Kern et al. (2015) found it important to use different densities for FYI and MYI, whereas
Kwok and Cunningham (2015) found the best correlation using only one density representative of FYI. Kwok and Cunningham (2015) also found the snow depth to be best represented by taking 70% of the Warren snow depth (Warren et al., 1999) over FYI as opposed to 50% used in Laxon et al. (2013)).

## 6 Future prospects: Expectations and hopes

This appears a very propitious time for Arctic studies, with many decades of data available enabling further honing of algo-
rithms, and the prospect of new satellite missions, in situ data and new initiatives to utilise multiple datasets synergetically. We consider, below, some of the areas in which further advances may be anticipated.

### 6.1 Improvements in processing

Although the launch of CryoSat-2 in 2010 heralded a new era in Arctic altimetry, with its high inclination orbit and finer resolution through SAR processing, there still remains much to be done with the preceding two decades of LRM altimetry.
Recent advances in data storage and computer processing power mean that many people now have the potential to process the entire ERS-1, ERS-2, Envisat data record, rather than relying on the space agencies. This has been greatly assisted by ESA's





REAPER project (Brockley et al., 2017), which has made the data from early missions more readily accessible, coupled with improved orbits and corrections.

A number of new processing strategies have been developed (Poisson et al., 2018; Passaro et al., 2018) which have yet to be applied to the full set of data on the 35-day ERS/Envisat orbit. Further improvements in retracking and quality control may be expected. Firstly this will encompass improved statistical approaches to select the different echo types, but potentially there may also be retracking algorithms that utilise the extra information in waveforms adjacent to a lead, rather than solely retracking the single return directly over it. Such an approach, gathering information from across the hyperbolic trajectory (see Fig. 12) could build on previous ideas for detecting individual bright targets (Gomez-Enri et al., 2010; Tournadre, 2007; Boergens et al., 2016).

## 6.2 New missions with new capabilities

Table 1 catalogued all the radar altimetry missions covering the Arctic beyond 72°N, including Sentinel-3B launched in April 2018. The recent Sentinel-3 missions not only offer the prospect of many decades further coverage through subsequent launches, but also offer the potential to improve the historical record. This is because early in their combined tandem mission observing the same ground track 30 seconds apart, one will operate as a SAR altimeter and the other in LRM mode. This will enable a much closer comparison of waveform classification and retracking in these two different modes. The understanding gained about their differential response to, say, small leads or melt ponds, can be used to develop a more consistent processing scheme to deliver the homogeneous datasets on Arctic sea level and freeboard that are required for climate studies.

The Envisat radar altimeter RA-2 not only worked at Ku-band but also at S-band, which was exploited for ice studies over Antarctica and Greenland (Legresy et al., 2005; Tran et al., 2008). However, this was only rarely used for ice classification over the Arctic (Tran et al., 2009), as the information from the passive MWR was more useful in that study than the S-band radar measurements. The potential advantages of dual-frequency altimetry may be better realised with the Sentinel-3 spacecraft, since their measurements (at C-band) have the same width sampling bins as at Ku-band (which was not the case for Envisat) and thus better recording of the waveforms at the secondary frequency. Figure 19 shows the mean difference between the backscatter strengths ($\sigma_{Ku}^0$ and $\sigma_C^0$) for three different months over the winter 2016/17. The chosen scaling for Sentinel-3 $\sigma^0$ values makes the mean values over the ocean similar (although there is some variation with wind conditions). The series of three plots shows the expanding area of sea ice (confirmed by the 50% SIC contour in pink), with the recently-formed ice having a $\sigma_{Ku}^0 - \sigma_C^0$ signature of ~1.5 to 2.5 dB, but the regions where ice has been present for several months showing negative values. This possibly points to some change in sea-ice properties over the first few months, but it does not provide a distinction between FYI and MYI. Work to support the use of these auxiliary measurements needs to be encouraged.

There are also plans for successor missions for both AltiKa and CryoSat-2. In particular, CryoSat-2 has been operating for 8 years, and there is a compelling case for another mission to observe the Arctic poleward of 81.5°N, as this regions contains most of the MYI. Future missions will not only help ensure the long-term recording of Arctic sea level and ice freeboard, but could also improve the snow depth climatology through the joint exploitation of Ku-band and Ka-band data (Guerreiro et al., 2017).



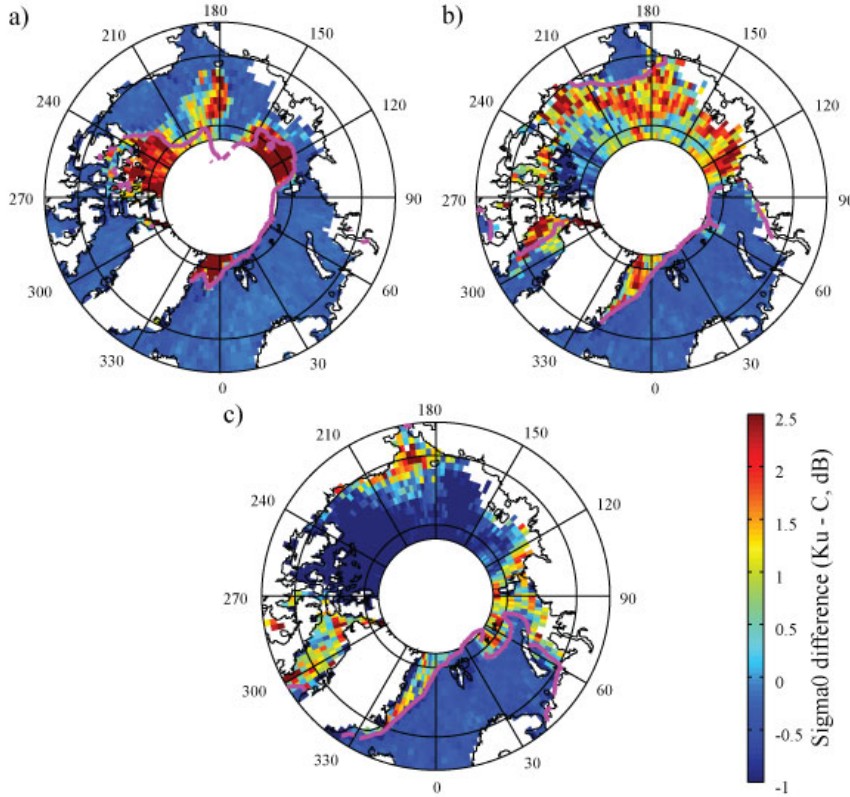

**Figure 19.** Mean difference in Sentinel-3A backscatter values at Ku-band and C-band, for a) Sept. 2016, b) Nov. 2016, c) Jan. 2017. The pink lines show the 50% sea ice concentration from SSM/I sensors (Cavalieri et al.) for the same months.

This review paper has focused on radar altimetry only. However, NASA's ICESat mission (Schutz et al., 2005) carried a laser altimeter that has been operated between February 2003 and October 2009, with the capability to retrieve freeboard (Kwok et al.). But in contrast to radar altimetry, laser altimetry is affected by clouds, and ICESat measurements were restricted to two periods per winter season (October/November and February/March). The combination of laser and radar altimtery is
5    challenging because of the very different footprint sizes, effect of clouds on laser measurements, and different characteristics of the penetration into the snow pack on top of the ice. However, the launch of ICESat-2 (scheduled for September 2018) should prompt renewed effort to combine radar and laser range measurements. The Advanced Topographic Laser Altimeter System (ATLAS) onboard ICESat-2 uses a multi-beam approach with 6 laser beams, arranged in 3 pairs pointed on the ground at intervals of 3.3 km across track (Markus et al., 2017). Coincident measurements of ICESat-2 and CryoSat-2 also have the
10    potential capability to estimate snow depth.



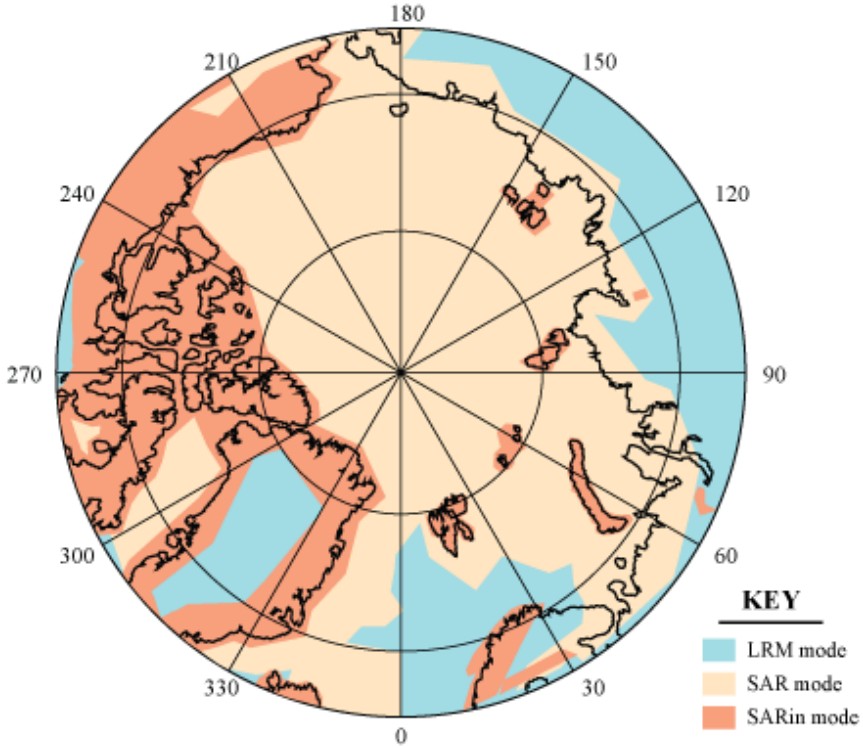

**Figure 20.** Mode mask v3.9 for CryoSat-2 operations: CryoSat-2 operates in SAR mode over most of the Arctic Ocean, with SARin mode over the Canadian archipelago and most Arctic islands and LRM mode over the Greenland plateau.

## 6.3 Realizing the potential of SAR interferometry (SARin)

CryoSat-2 has pioneered two advances in altimetry. As well as being the first spaceborne system to use SAR altimetry, it has a second antenna 1.2 m across track from the one that both transmits and receives signals. This gives it an interferometric capability, as the phase of the signals received by the two antennas can be compared to determine the off-nadir location of the signal (Wingham et al., 2004). The original purpose was to map the exact location of the return signal in areas with highly varying topography such as glaciers, smaller ice caps, and margins of ice sheets (Wingham et al., 2006) (see Fig. 20), but it has also been shown to improve the retrieval of sea surface heights in coastal (Abulaitijiang et al., 2015; García et al., 2018) and sea-ice-covered regions (Armitage and Davidson, 2014), thus reducing the uncertainty of sea-ice freeboard heights (Bella et al., 2018).

As described in Section 3.4.1 snagging leads to underestimates in the sea surface height due to locking on bright off-nadir targets. This highly affects classical altimetry observations due to the large footprints and even affects CryoSat-2 in SAR-mode by recording bright targets that are up to 13 km in the across-track direction (Abulaitijiang et al., 2015; Beckers et al.,





2013) which is outside the nominal across-track footprint size. Such snagging events are normally circumvented by suitable retracking and discarding waveforms including reflections from off-nadir leads.

Usually, sea surface height and sea-ice freeboard in SARIn areas are processed using a SAR-like approach (Tilling et al., 2018; Idžanović et al., 2017) with degraded noise level compared with the real SAR acquisition due to the lower burst repetition

frequency of the SARIn mode (Wingham et al., 2006). By using the phase information from the SARIn mode to range correct off-nadir leads, the accuracy and precision of the estimated sea surface height is improved by increasing the number of valid waveforms despite the degraded noise level (Armitage and Davidson, 2014). The inclusion of the increased number of retrieved sea surface height estimates (with only ∼35% of the discarded waveforms of the SAR-like case) results in a reduction of the total random freeboard uncertainty of 40% (Bella et al., 2018).

The SARIn capabilities are currently a unique feature of CryoSat-2 altimeter mission, and restricted to specified area of complex topography. However, the launch of NASA/CNES Surface Water and Ocean Topography (SWOT) mission in 2021 will provide Ka-band SARIn altimetry globally between 78°S and 78°N. To support further investigation of the advantages and limitations of SARIn altimetry, most recent ESA CryoVEx campaigns in 2017/2018 have collected airborne Ka-band altimetry data in SARIn mode in the Arctic and Antarctic (see also Section 5.2) along selected CryoSat-2, Sentinel-3A and

AltiKa ground tracks (Skourup et al., 2017). This was epitomised by a coordinated sea-ice flight involving for the first time four aircraft, carrying a suite of instruments to monitor the snow and sea ice along a CryoSat-2 ground track.

### 6.4    Utilising data fusion techniques

Due to the sensor and orbit characteristics, satellite retrievals of sea-ice thickness differ in spatial and temporal resolution as well as in the sensitivity to certain sea-ice types and thickness ranges. The aim of satellite data fusion is to take advantage of

the complementarity of retrievals derived from different satellite sensors.

One of the major objectives of the CryoSat-2 radar altimeter mission is the retrieval of Arctic sea-ice thickness. It was designed to observe thick first-year and perennial sea ice, but it has a larger relative uncertainty for thin seasonal sea ice. CryoSat-2 uncertainties contain contributions that are associated with speckle noise, sea surface height estimation, snow depth, and densities of ice and snow (Ricker et al., 2014).

On the other hand, the 1.4 GHz (L-band) radiometer on the SMOS (Soil Moisture and Ocean Salinity) satellite has been used successfully to retrieve the thickness of thin ice in the marginal ice zone and during the freeze-up (Kaleschke et al.). The method is based on analyzing the surface brightness temperatures using a thickness-dependent emission model, and the overall uncertainty contains contributions from the errors in measured brightness temperatures, the uncertainty in sea-ice salinity and temperature, as well as the assumptions for radiation and thermodynamic models (Tian-Kunze et al., 2014).

Figure 21 shows the relative uncertainties (as calculated in (Ricker et al., 2017)) for CryoSat-2 and SMOS monthly means for the winter season 2013/2014. While the SMOS uncertainties are low over thin ice (< 1 m), SMOS's sensitivity for thicker ice (>1 m) is limited and thicknesses above 1.5 m are not retrieved. In contrast, the absolute uncertainty of CryoSat-2 estimates over thick ice is the same or less than that over thin ice, so the relative uncertainty drops with increasing sea-ice thickness.





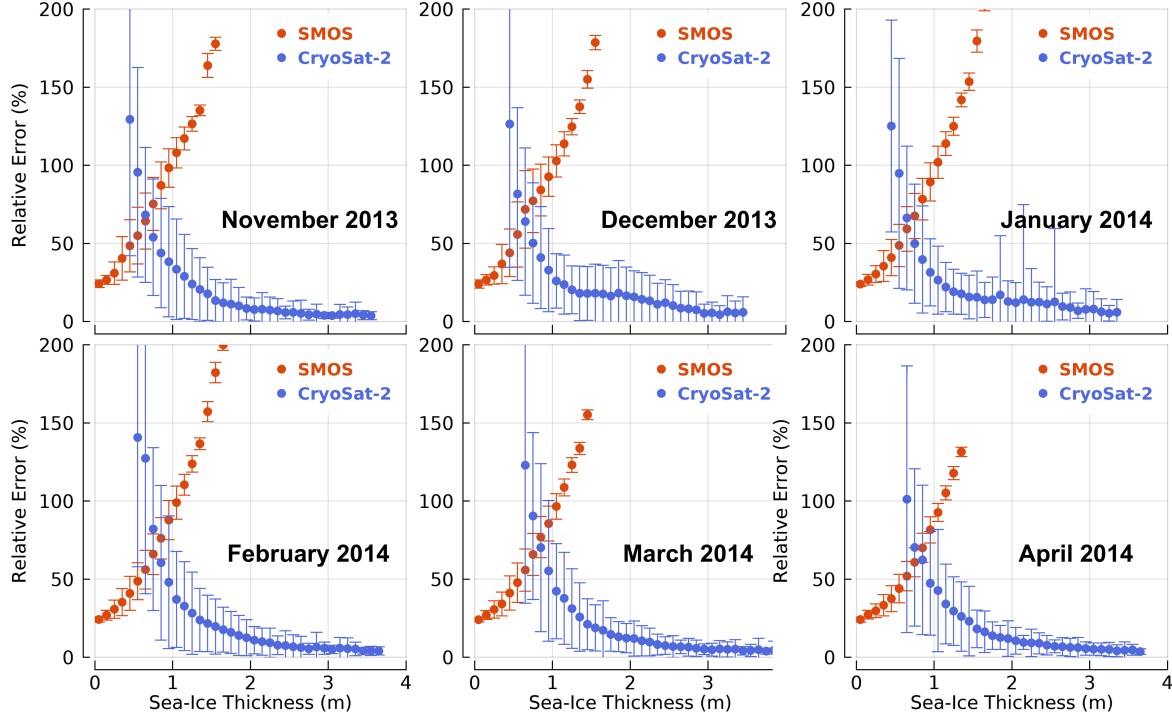

**Figure 21.** Relative uncertainties for CryoSat-2 and SMOS Arctic ice thickness estimates during one winter season, November-April, 2013/2014. Relative uncertainties are binned in steps of 0.25 m of sea-ice thickness. Error bars represent the standard deviation of the mean relative uncertainty for each bin.

Ricker et al. (2017) developed a method of completing and improving Arctic sea-ice thickness information by merging CryoSat-2 and SMOS sea-ice thickness retrievals based on an optimal interpolation scheme. The merged product overcomes several issues associated with single-mission retrievals, and provides a more accurate and comprehensive view on the state of Arctic sea-ice thickness. This approach can be adopted for recently launched altimeter missions such as Sentinel-3A and 5 Sentinel-3B. Their orbital inclination (see Table 1) results in a larger pole hole (region of no altimeter observations), but on the other hand, the density of Sentinel-3 orbits at 81.4°N is much higher than for CryoSat-2, which leads to a synergetic effect when both missions are combined. However, the continued operation of an altimeter in a CryoSat-like orbit is essential for monitoring sea-ice thickness north of the turning latitude of the Sentinel-3 missions.

## 6.5 Enhanced in situ observations

10 In the derivation of freeboard from altimetry data, a major source of error is the uncertainties in the various physical constants used in the calculation (see Eq. 2). The snow depth climatology is from a study by Warren et al. (1999), which is based on in situ observations obtained during 1954-91 and is no longer representative of the current snow depth conditions (Kurtz et al., 2013). For sea-ice density, most studies use the measurements provided by Alexandrov et al. (2010), which are not just outdated





(1928-93), but are also restricted to the coastal Siberian regions and are therefore not representative of the entire Arctic basin. In this context, new measurements of sea-ice density and snow depth should be obtained at basin scale in order to update the current parametrizations used to convert freeboard height to ice thickness.

A major advance is likely to come from the MOSAIC programme (MOSAIC, 2018), in which the RV PolarStern will act as
the central observation site for am international 12-month multi-disciplinary study that will encompass measuring snow depth and morphology, as well as ice draft, on a variety of scales. In particular there will be surveys on at least a weekly basis at a number of sites spanning different sea-ice conditions, which will include the development of melt ponds during the thaw cycle. Such a large programme will not only furnish a better parameterization of the factors affecting these constants, but will also create a useful database of validation data. However, even such a campaign, which will be lodged within FYI, cannot
fully address the diversity of conditions across the whole Arctic, and so contributions from other drifting stations, icebreaker missions and dedicated cal/val exercises (e.g. associated with ICESat-2) will still be essential.

## 7  Conclusions

This paper has provided a review of the technical aspects of radar altimetry over the Arctic for both sea level and sea-ice studies, and complements the scientific review provided by Johannessen and Andersen (2017). It has shown how surface type
affects the shape and strength of the return waveforms, both for LRM and SAR altimeters. It has covered the challenges of robustly classifying the waveforms (and also of assessing such a classification), and then developing retracking approaches for deriving the height of the reflecting surface, especially with spurious extra signals due to off-nadir bright targets.

A persistent issue is that of reliably classifying data according to surface type. Whilst there are a plurality of solutions (with many just having a visual validation for a selected scene), there have been few papers that compared several methods
quantitatively against reliable independent datasets. Even the statistical or machine-learning approaches typically rely on some subjective operator classifications as their reference. Effort is required to collate all the available ground truth data (whether in situ or from airborne or spaceborne sensors) that coincide with satellite altimetry. Open access to such an extensive database, plus protocols for systematic validation, would allow new approaches to be more reliably benchmarked. For the geophysical products (principally sea level and sea ice thickness) further developments in the interpolation and mapping may be required
that better accommodate the short-correlation lengths and the non-synoptic nature of each month's observations.

Whilst we have the benefit of 25+ years of Arctic altimetry data, the accurate estimation of potential climate-related change is still very challenging, because of the need to understand the differences between all the altimeters. ESA's long-term vision for 15 years of Sentinel-3 altimetry provides some confidence that the majority of the Arctic will continue to be monitored, but there is a potential gap in the coverage north of 81.5°N if CryoSat-2 ceases operation. The need for in situ measurements
remains strong, not just for the purpose of satellite validation, but also to provide a better understanding of the factors (snow depth, melt ponds, refreezing of leads) that affect the altimeter's waveforms.




**Table 3.** List of Acronyms used

| | | | |
|---|---|---|---|
| ADCP | Acoustic Doppler Current Profiler | mss | mean square slope |
| AEM | : Airborne ElectroMagnetic sensor | MSS | Mean Sea Surface |
| AGC | Automatic Gain Control | MYI | Multi-Year Ice |
| ALES | Adaptive Leading Edge Sub-waveform | OCOG | Offset Centre-Of-Gravity |
| BGEP | Beaufort Gyre Exploration Project | OIB | Operation IceBridge |
| BH | Brown-Hayne | PLRM | Pseudo Low-Rate Measurement |
| CCI | Climate Change Initiative | PP | Pulse Peakiness |
| CryoVEx | Cryosat Validation Experiment | PTR | Point Target Response |
| DAC | Dynamic Atmosphere Correction | RA | Radar Altimeter |
| DTC | Dry Tropospheric Correction | RIP | Range-Integrated Power |
| ERS | European Remote-sensing Satellite | SAR | Synthetic Aperture Radar |
| Envisat | Environmental Satellite | SK | Stack Kurtosis |
| ESA | European Space Agency | SLA | Sea level anomaly |
| FYI | First Year Ice | SMOS | Soil Moisture and Ocean Salinity |
| GDR | Geophysical Data Record | SP | Stack Peakiness |
| GOCE | Gravity field and steady-state Ocean | SS | Stack Skewness |
| | Circulation Explorer | SSD | Stack Standard Deviation |
| GRACE | Gravity Recovery and Climate Experiment | TES | Trailing Edge Slope |
| ICESat | Ice, Cloud,and land Elevation Satellite | TFMRA | Threshold First Maximum Retracker Algorithm |
| IMB | Ice Mass Balance | ULS | Upward-Looking Sonar |
| LEW | Leading Edge Width | WTC | Wet Tropospheric Correction |
| LRM | Low Rate Measurement | | |
| MDT | Mean Dynamic Topography | | |
| MP | Maximal Power | | |

*Author contributions.* This review article was led and co-ordinated by G.Q. with contributions and internal review by all named authors.

*Competing interests.* The authors declare no conflict of interest. The European Space Agency, who funded the CCI work, retained an active interest in the development of the work through their Technical Officers, Jerome Benveniste and Pascal Lecomte, but they had no role in the writing of this manuscript.

5 *Acknowledgements.* The authors were originally supported by the European Space Agency for work towards their Climate Change Initiative programme (Sea Level CCI and Sea Ice CCI). The opportunity for cross-disciplinary discussion was assisted by two meetings funded by the



International Space Science Institute. We acknowledge the contributions to those discussions made by Pierre Thibaut and Jean-Christophe Poisson and are sad that they were unable to contribute to this paper. GQ is grateful for the financial support of ESA through the Sentinel-3 Mission Performance Centre, and to Francesco Nencioli for help with the LaTeX formatting. Data on the climatolgical sea-ice extent were provided by the National Snow and Ice Data Center (NSIDC).



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
