# Peer review of "Review of Radar Altimetry Techniques over the Arctic Ocean: Recent Progress and Future Opportunities for Sea Level and Sea Ice Research"

_The Cryosphere, 2018_

## Referee Comment (RC1) · Anonymous Referee #1 · 9 Aug 2018

General Comments:

The authors aim to provide a review of radar altimetry techniques over the Arctic Ocean for retrievals of sea surface height and sea ice freeboard/thickness. The scope is broad and includes a discussion of the available data sets, certain altimetry basics, altimeters, surface classification, retracking techniques, validation approaches, needed geophysical corrections for retrievals, and future opportunities. This is rather ambitious undertaking in attempting to address both the sea surface and ice surface derivations at the same time.

I find the manuscript to be somewhat disappointing; there are a number of technical inconsistencies and important omissions. I also expected a review rather than a survey. In a review, I should expect the authors to provide insights and critical/quantitative assessments of the current state of knowledge, what has been achieved, and what needs to happen to push forward. These essential elements are missing. Instead, I find a broad-brush survey that is not quite comprehensive enough for the non-expert (terms are insufficiently defined) and not that useful for the expert.

The manuscript is rather qualitative and offers few quantitative insights; the authors do not describe what is required (in terms of altimetric accuracy) to understand sea surface anomalies, and sea ice thickness, and the various error sources. The introduction should first detail the centimeter-level requirements for understanding changes in, for example, geostrophic circulation and sea ice thickness. The remainder of the manuscript should then answer questions relevant to what has been achieved (quantitatively, i.e., data quality) in recent altimetry work, geophysical understanding of trends or variability (in the published literature), and recommendations for future work.

The authors surveyed a number of techniques on re-tracking and approaches for validation of the results. However, the authors supplied very little conclusive evidence that there have been progress because the difficulty in assessing the variety of results and approaches. One of the few places I found a quantitative discussion of data quality was in the section on ice thickness.

Snow depth, one of the sources of uncertainty in ice thickness retrieval, received relatively little attention. Also, the need to understand snow properties, including salinity and stratigraphy, rather than just density and depth should be emphasized. Salinity, especially, affects freeboard estimates especially if a dual-band altimeter were to be used for snow depth retrieval.

The conclusions section should provide a summary of what has been accomplished in the several decades of radar altimetry over the Arctic. What was offered in the second paragraph of the conclusions section seems to suggest that retracking is the primary concern, and perhaps it is for altimetry experts. But, understanding the geophysical medium being sensed, which controls the returns and therefore the quality of geophysical retrievals, is as important if not more so. This aspect is missing and should be emphasized.

I do not recommend publication in its present form because a more focused manuscript with critical/quantitative assessments of procedures and current geophysical utility of radar altimetry would entail significant revisions to this paper. Any revisions to the current paper to address the above concerns should be considered a new submission and reviewed as such.

Detailed Comments:

Page.Line number

2.10 'ocean' temperature is perhaps better than 'sea'.

2.21: For navigational purposes, I think ice thickness would be more of an interest than ice volume.

2.25: Are ridging rates really a requirement? Please provide a citation here.

2.27: "Daily evolutions and high spatial resolution…" of what?

Figure 1. I question the usefulness of this figure, especially the bathymetry bit because it doesn't seem relevant to the review at hand. Also, it's difficult to see the purple circulation patterns.

4.4: '…typically every 300 m…' is not correct especially with the altimeters in Fig. 2. One needs synthetic aperture processing to achieve resolutions of 300-m along-track– showing the spot size of each of these altimeters in Table 1 would be useful. Since this is a review article, it may also be important to explain how these spot sizes, perhaps briefly, are defined.

4.5: The connection between along-track sampling and spatial coverage seems odd: "… tens of days of along-track sampling are required… to give good spatial sampling…' I suppose you mean across-track separation between the orbits.

4.15 I don't think the altimeters are not in the same orbit, only same orbit inclination.

Table 1. 'Non-circular antenna…" is not correct but why is this significance here?

5.1 '…superposition of multiple…" perhaps better to say aperture synthesis from multiple viewing geometries.

5.6: This may be a good place to provide a better definition of LRM.

5.9: What are "continuous ice floes"?

8.29: I think there needs to be a justification as to why these two techniques were reviewed in this manuscript. Do the authors think that these are more promising than others?

11.9: 'narrower' is better than 'smaller'.

11.10: please list the three techniques.

12.5: I guess the point to note is that the techniques are typically tuned to specific altimeters.

12.11: It would be useful, in a review article, to say something about actual lead widths in this context.

13.5: I think you mean ice drift and not currents. Also, 100 m hr-1 is not a good average number to use for the Arctic – more useful to provide a range.

13.10: what is water-leaving radiance?

13.35: this doesn't bode well for the use of SAR imagery. Dark areas in SAR imagery are not necessarily leads (meaning open leads); they could be quite thick first-year ice.

Section 2: Thus far, the use of other data sets to validate surface classification seems quite discouraging, wouldn't you say?

17.2: how does the thermal noise in the altimeter translate into ranging noise?

Figure 14. The area over which these anomalies are calculated should be shown.

26.1: The different geophysical processes that contribute to sea surface heights should be described and why they need to be removed for estimation of dynamic ocean topography and the calculation of geostrophic current should be clarified.

26.15 The distinction between DAC and inverted barometer should be described.

Section 4.1: discussion of a number of tides is missing, e.g., solid earth, pole, etc. Even though the authors chose to discuss two that are important, the others should be mentioned. I think the magnitude of these tides in the Arctic should be listed.

28.10: The length scale over which residual ocean tilt (due to eddies and other circulation patterns) may affect freeboard calculations is important and should be discussed.

28 – The current developments in estimation of snow depth should receive more attention.

32.30  The authors failed to mention the potential effect of salinity on returns – due to brine wicking-  from the snow-ice boundary (reported in Nandan et al., 2018). This should be discussed.

Section 6.4 seems to be outside of scope of radar altimetry.

40.1: It should be snow properties, including salinity and stratigraphy, rather than just density and depth.

Section 7: The second paragraph suggests that retracking is the primary concern, and perhaps it is for altimetry experts. But, understanding the geophysical medium being sensed, which controls the returns and therefore the geophysical retrievals, is as important if not more so. This aspect should be emphasized.

---

## Referee Comment (RC2) · Anonymous Referee #2 · 10 Aug 2018

This paper presents a broad review of radar altimetry for the Arctic Ocean. While I thought many of the topics were relevant and needed for a review paper, I felt that in general the paper does not demonstrate or present a sufficient depth of understanding of the subject as there are many poorly worded statements and factual errors present throughout. Furthermore, I felt that the manuscript presented an overt subjective bias towards the authors' works rather than presenting a more objective review of the subject in general. Specific comments on these aspects are noted below.

I also agree with much of what was stated in the previous review and would like to see

a significant revision addressing those points as well as the ones raised here prior to publication.

P2 L9-11: This statement about the current year warming of Greenland seems a bit out of context in a review paper like this.

P2 L13: State that this is an expectation of a summer ice-free Arctic Ocean, not year-round.

Figure 1b) Check that the grey circle area is correct, I think the SSM/I pole hole is much smaller than this.

P4 L3: The sampling resolution is different than the footprint size, it would be useful to distinguish that here.

P4 L17 and throughout: I think LRM may have been a term adopted more specifically for CryoSat-2, I would say "pulse limited" here instead.

Table 1: The antenna pattern of CryoSat-2 is non-circular, not the antenna itself.

P5 L2: I'm not sure what is meant by "impulse-like shape". By definition the impulse response of an altimeter takes into account geometric spreading, so I don't think Figure 3 shows this aspect. The SAR processed waveforms do indeed have a much faster decay time than returns from only pulse-limited systems though.

P6 L6: It's not just a smooth curve that is fitted to the return, as described in the cited references it is actually a mathematical function which takes into account the instrument impulse response, point target response, and surface roughness.

P6 L10: I think the use of reflectivity here is incorrect. Do you mean the reflection from the Fresnel coefficient? That is not what leads to the range of different sigma_0 values, rather it is the wide variations in surface roughness.

P6 L10-14: This effectively ignores why the returns from leads are confined to a small number of range bins. It also needs to be made clear that the leads represent a small

spatial area which is what leads to the return being much more like that from the point target response.

Figure 4 b): I'm not sure this is a fair comparison between Envisat and AltiKA since the y-axis is in units of instrument counts, unless the conversion between instrument counts and power is the same between both satellites.

P8 L17-18: I don't think it is true that the actual thresholds are unimportant. For example, Armitage and Davidson, 2014 found that the value of pulse peakiness used had a relationship with the bias in retrieved sea surface height.

Figure 6: What is Brownian + pic?

P10 L16: I don't believe the Doppler shift of the echoes are explicitly recorded, but are rather used within the processing loop itself. See Wingham et al., 2006.

P11 L8-9: I don't think this statement is true. Since even small leads have a much higher backscatter than the surrounding ice floes they tend to dominate the echoes from both SAR and traditional pulse limited systems. However, the SAR processing limits the contribution from off-nadir leads in the along track direction.

Figure 7: I'm unsure what this figure is showing. The method used for the threshold on the stack peakiness is unclear as no threshold is specified. Additionally it lacks context, how does it compare to previously published methods?

P12 L5: I think more importantly it is distinguishing between returns from melt ponds and returns from leads, not just recognizing melt ponds.

P 13 L6-7: I'm not sure what is meant here "...have been manually edited for the effects of land and clouds, with no automation of the processing."

P13 second paragraph: I think this paragraph lacks a good description of what is important for radar altimetry discrimination of surface types.

P13 L32-33: Any smooth surface can produce a low backscatter value for a SAR instrument, the surface could simply be very smooth first year ice and is not necessarily a lead.

Section 2.3.2: This section focuses much on validation of surface classifications of a few select algorithms, but I am quite surprised it lacks comparison with that used by Laxon et al in various studies as these are widely employed in the standard ESA processing for CryoSat-2 and have a long history of use going back to ERS and Envisat.

P16 L10: I think the written out acronym here is wrong.

P16 L13-14: These studies did not use the imagery data in a comparative sense as stated here.

P16 L18-25: This paragraph is poorly written and carries a negative tone towards the cited studies. The King reference is lacking a year, and I don't think the Connor et al., 2009 citation accurately represents what was shown in the study.

P16 L31-32: This is a poorly written sentence.

P17 L9-11: This statement is not true, the returns from sea ice are quite a bit different than open ocean returns most prominently they tend to have a much shorter trailing edge, but this is not due to snow volume scattering.

P17 L17-19: The Giles et al., 2007 paper did not use an OCOG retracker. I think in general this section is missing much of the historical work done by Seymour Laxon in retracking of floes.

P18 L1-7: This section clearly misrepresents the historical work done towards development of CryoSat-2 retracking. The widely cited Laxon et al., 2013 study utilized a threshold tracker for sea ice floes and did not use an OCOG tracker. This threshold tracker superseded the TFMRA tracker which is quite similar on a conceptual level.

P18 L 9-11: I'm not sure what this has to do with how snow impacts the results?

P18 L13-14: I think it is quite a subjective statement on the author's part that TFMRA

is the most widely used tracker for SAR data and should be removed. I think the statement also does a disservice to the historical development of sea ice retrackers in general by framing this in such a way, and conveys a bias towards the authors' own work.

Section 3.1.3: I find this section to be poorly written. The second paragraph inappropriately mixes in a discussion of models for ice and open ocean returns even though it has been quite clearly shown in prior studies that the models for open ocean returns do not work for sea ice regions. The section also ignores what physical modeling of ice returns have shown with describing variations in the radar returns as elaborated on in previous sections.

P18 L30: The Rivas et al., 2006 paper did not compare sea ice and open ocean roughness so this sentence needs some revision.

P19 L9-10: How is the waveform "severely undersampled" for sea ice leads? The range resolution sampling is set by the bandwidth of the altimeter. So it would seem to me more appropriate to discuss this in the context of the need for high accuracy or precision in the data which is limited by the bandwidth of the satellite altimeters.

P19 L11-15: Here I think it is missed that the work of Laxon et al., 2013 used the tracker from Giles et al., 2007 and this is used in the CryoSat-2 sea ice product. Again, this paragraph conveys a bias towards the authors' own work.

P19 L20-26: This paragraph ignores the fact that the range resolution is determined by the received bandwidth, zero padding of the waveform does not actually confer a real increase to the range resolution, and associating this to the calibration functions used to obtain mm level path delays is highly inappropriate. Zero padding can be useful in certain algorithm approaches or in visualizing the data, but this needs to be more clearly written.

P19 L29-30: Here I think the authors miss an important need for zero padding of SAR

waveforms as demonstrated by the cited Jenson, 1999 study.

Section 3.3: The cited study by Dinardo et al., 2017 did not demonstrate retrievals over sea ice or leads, but was focused on the coasts of German Bight and West Baltic Sea. So it seems not credible to put the claim in this review paper that the methodology was successfully demonstrated to retrieve sea ice freeboard from CryoSat-2 SAR data.

I think here the authors also miss discussing the Kurtz et al., 2014 study which showed a unified physical tracker for both sea ice leads and floes from SAR waveforms.

Again, this seems to show clear biases towards the authors' own work.

Section 3.4.1: Here the authors miss the opportunity to more thoroughly describe the capability of CryoSat-2 to determine phase and thus estimate the range bias present over sea ice as shown in Armitage and Davidson.

P23 L11: This sentence is not accurate: "This is due to the along-track beam-limited resolution, which also reduces the backscatter from across-track points." The SAR processing does not impact the backscatter from across-track points in this way.

P25 L4: The western Greenland Sea is quite often ice-covered.

Section 4.1.1: Here it would be useful to state the magnitude and variability of the corrections.

P29 L8-9: Not all studies use a 50% reduction for snow climatology, the Kwok and Cunningham study used a different value.

Section 4.2.2: I think here the important factor of snow depth and density in leading to a lower propagation speed of light in the snow is missed.

P32 L18, P34 L4-5: From the text it seems unclear what is measured by Operation Ice-Bridge, in my reading it seems to imply only laser altimeter freeboard. But it should be noted that snow depth measurements from radar altimetry are also used to determine snow.

---

## Referee Comment (RC3) · Anonymous Referee #3 · 22 Aug 2018

The paper 'Review of Radar Altimetry Techniques over the Arctic Ocean: Recent Progress and Future Opportunities for Sea Level and Sea Ice Research by Quartly et al. aim to provide an overview and review of various radar altimetry techniques over the Arctic Ocean for sea ice freeboard/thickness and sea surface height retrievals. The paper reviews altimetry techniques in a very broad context and includes a comprehensive review of altimetric data sets, basic-to-advanced concepts of radar altimetry, retracking algorithms, and geophysical correction methods for unambiguous parameter retrievals.

Since I am an altimetry expert on sea ice freeboard/thickness retrievals, I would want to concentrate my focus on reviewing the sea ice side of this paper. Overall, I find this manuscript not suitable for publication at this point of time, as the authors fail to provide a comprehensive treatment snow depth and its associated thermal and geophysical properties as a major source of uncertainty in sea ice freeboard/thickness retrievals. The authors should understand the impact of critical snow geophysical properties such as snow temperature, snow salinity, snow grain microstructure, in addition to snow thickness and density (which the authors have already pointed out as error sources in this manuscript). This is to be emphasized in more detail as snow covers on sea ice is still considered to be the greatest source of uncertainty, with currently used snow thickness and density assumptions considered to be unassumingly vague. A critical assessment of how snow covers and its geophysical properties impact sea ice thickness freeboard/thickness should be accounted for in this manuscript and requires substantial focus in this manuscript. I would recommend you reviewing latest literature on how snow thickness (Tonboe et al., 2006; Tonboe et al., 2010; Ricker et al., 2014, 2015, 2017), snow density (Alexandrov et al., 2010, Tonboe et al., 2006), snow temperature (Willatt et al., 2011), snow compression (Tilling et al., 2015; Kern et al., 2015), snow salinity (Nandan et al., 2017; Nandan et al., 2016) impact freeboard retrievals from radar altimetry.

I also agree with the comments from the other two reviewers and would like to see a more detailed revision addressing the points above and also from the other reviewers. Hence, I do not recommend publication in its present stage, and would recommend to resubmit this manuscript in a revised form. I will be happy to provide specific comments to the revised manuscript, once the general comments are addressed.

---

## Author Comment (AC1) · 23 Oct 2018

**General Response**

Writing a review of radar altimetry techniques over the Arctic may have been ambitious (as noted by Rev. 1), but we felt that there was a great need (as shown by the number of downloads) for somebody to do this.  We accept that in our planning of this paper we omitted to discuss the key attributes of snow on ice and their effect; this is now being rectified with a few paragraphs.  Our intention with a "review" was to provide a brief synopsis of the different steps needed and a light discussion of the alternative procedures; we did not aim to fully quantify the performance of  each procedure and define an optimal  processing chain, because there is clearly no consensus on all the parts.

**Response to Reviewer 1**

Review of paper by Quartly et al.

**General Comments:**

The authors aim to provide a review of radar altimetry techniques over the Arctic Ocean for retrievals of sea surface height and sea ice freeboard/thickness. The scope is broad and includes a discussion of the available data sets, certain altimetry basics, altimeters, surface classification, retracking techniques, validation approaches, needed geophysical corrections for retrievals, and future opportunities. This is rather ambitious undertaking in attempting to address both the sea surface and ice surface derivations at the same time.

I find the manuscript to be somewhat disappointing; there are a number of technical inconsistencies and important omissions. I also expected a review rather than a survey. In a review, I should expect the authors to provide insights and critical/quantitative assessments of the current state of knowledge, what has been achieved, and what needs to happen to push forward. These essential elements are missing. Instead, I find a broadbrush survey that is not quite comprehensive enough for the non-expert (terms are insufficiently defined) and not that useful for the expert.
This "review" was intended to cover all aspects sufficiently to enable the non-expert user to understand the key issues, rather than quantitatively compare lots of methods of discrimination, waveform retracking, freeboard calculation etc.  If there were omissions or lack of definition of terms, we have amended where we have received specific details from the reviewers.

The manuscript is rather qualitative and offers few quantitative insights; the authors do not describe what is required (in terms of altimetric accuracy) to understand sea surface anomalies, and sea ice thickness, and the various error sources. The introduction should first detail the centimeter-level requirements for understanding changes in, for example, geostrophic circulation and sea ice thickness. The remainder of the manuscript should then answer questions relevant to what has been achieved (quantitatively, i.e., data quality) in

recent altimetry work, geophysical understanding of trends or variability (in the published literature), and recommendations for future work.

We agree that more quantitative information was needed when detailing the requirements, and have amended section 1.1 to reflect this, citing the GCOS measurement requirements. We do not aim to go on to "geophysical understanding of trends or variability" as that itself is a large topic and covered by the complementary work by Johannessen and Andersen.

The authors surveyed a number of techniques on re-tracking and approaches for validation of the results. However, the authors supplied very little conclusive evidence that there have been progress because the difficulty in assessing the variety of results and approaches. One of the few places I found a quantitative discussion of data quality was in the section on ice thickness.

Polar radar altimetry is a very dynamic field of research and some of the new techniques in processing radar acquisitions presented here have only being published over the last few years. Often these new methods are tested on case studies and have not yet been implemented in pan-Arctic fully operational processing chains. This makes a systematic comparison between methods challenging, and beyond the aims of this paper.

Snow depth, one of the sources of uncertainty in ice thickness retrieval, received relatively little attention. Also, the need to understand snow properties, including salinity and stratigraphy, rather than just density and depth should be emphasized. Salinity, specially, affects freeboard estimates especially if a dual-band altimeter were to be used for snow depth retrieval.

We have discussed the impact of snow depth on the freeboard and thickness retrievals (see sections 4.2.2, 5.2.1, 5.2.2), showing that snow on sea ice is a major source of uncertainty. But we agree with the reviewer that the effects of snow grain microstructure and salinity are not well represented in these sections. We have therefore added a few paragraphs discussing this in more detail.

The conclusions section should provide a summary of what has been accomplished in the several decades of radar altimetry over the Arctic. What was offered in the second paragraph of the conclusions section seems to suggest that retracking is the primary concern, and perhaps it is for altimetry experts. But, understanding the geophysical medium being sensed, which controls the returns and therefore the quality of geophysical retrievals, is as important if not more so. This aspect is missing and should be emphasized.

We have emphasised the importance of the surface type and its impact on the waveforms, but agree that the impact of properties of the sensed medium are slightly missed out in the conclusion. We have added some clarifications

I do not recommend publication in its present form because a more focused manuscript with critical/quantitative assessments of procedures and current geophysical utility of radar altimetry would entail significant revisions to this paper. Any revisions to the current paper to address the above concerns should be considered a new submission and reviewed as such.

Responding to the three reviewers certainly counts as "significant revision"; however, having completed most of that, we believe that we have a paper suitable for resubmission.

**Detailed Comments:**

Page.Line number

2.10 'ocean' temperature is perhaps better than 'sea'.

OK, "ocean"' has been put instead of "sea"

2.21: For navigational purposes, I think ice thickness would be more of an interest than ice volume.

"Volume" replaced by "thickness"

2.25: Are ridging rates really a requirement? Please provide a citation here.

"Ridging rates" replaced by ridged ice fraction. Consolidated sea ice features and their ridge fraction pose a hazard to both offshore structures and Arctic dwelling (as documented in ISO 19906: Petroleum and natural gas industries—Arctic offshore structures 2010).

2.27: "Daily evolutions and high spatial resolution…" of what?

Of previously mentioned parameters.  The text "For many of these parameters" is now inserted at start of sentence to make it absolutely clear.

Figure 1. I question the usefulness of this figure, especially the bathymetry bit because it doesn't seem relevant to the review at hand. Also, it's difficult to see the purple circulation patterns.

Figure has been redrawn to allow the circulation patterns to be seen more easily, and boxes added to lower disk to show the regions discussed in Fig. 14.

4.4: '…typically every 300 m…' is not correct especially with the altimeters in Fig. 2. One needs synthetic aperture processing to achieve resolutions of 300-m along-track– showing the spot size of each of these altimeters in Table 1 would be useful. Since this is a review article, it may also be important to explain how these spot sizes, perhaps briefly, are defined.

The text written was correct, but obviously misinterpreted by the reviewer, and the spot sizes were already explained in the text below Table 1.  However, the text "typically every 300 m" has been removed to avoid confusion.  Text has been added a few paragraphs later to clearly distinguish between posting rate and actual resolution of the measurements.

4.5: The connection between along-track sampling and spatial coverage seems odd: "… tens of days of along-track sampling are required… to give good spatial sampling…' I suppose you mean across-track separation between the orbits.

We agree that it is not clear what was meant by "good spatial sampling"; this has been reworded to "adequate spatial coverage is only achieved upon collating a pattern of different tracks spanning tens of days".

4.15 I don't think the altimeters are not in the same orbit, only same orbit inclination.

They are in the same orbit, which was a deliberate design so that the mean sea surface would be accurately known for the ground track to be occupied by AltiKa; it's just that the local time of the ascending node is different.  See these webpages for confirmation::
https://www.aviso.altimetry.fr/en/data/tools/pass-locator.html

https://directory.eoportal.org/web/eoportal/satellite-missions/s/saral

Table 1. 'Non-circular antenna…" is not correct but why is this significance here?
A number of webpages, including ESA's official Cryosat pages state that antenna is non-circular e.g. "They are slightly elliptical in outline" on http://www.esa.int/esapub/bulletin/bulletin122/bul122c_ratier.pdf   and Table 1 of https://geodesy.geology.ohio-state.edu/course/refpapers/Wingham_ASR_Cryosat06.pdf
"Non-circular" was mentioned in the Comments column just to make it absolutely clear that the two values in the beamwidth column should both be there; however now changed to "slightly elliptical" to avoid any reader thinking that they are rectangular!

5.1 '…superposition of multiple…" perhaps better to say aperture synthesis from multiple viewing geometries.
We have rephrased this as "due to the accumulation of looks (multilooking) from multiple viewing geometries."

5.6: This may be a good place to provide a better definition of LRM.
[This comment actually referred to 6.5]  A little text has been added to define LRM as Low-Resolution Measurement and also describe it s "conventional" -- references are now provided to show the change in data & processing associated with the change in concept from LRM->SARM

5.9: What are "continuous ice floes"?
[This comment actually referred to 6.9]  We meant a surface for which there was negligible signal from leads.  This is now reworded as "sea ice free from leads"".

8.29: I think there needs to be a justification as to why these two techniques were reviewed in this manuscript. Do the authors think that these are more promising than others?
We now provide some text explaining that these two techniques are examples of machine-learning techniques that use i) supervised and ii) unsupervised classification, and also refer to a recent paper by Dettmerring et al (2018), which provides more information on such methods.

11.9: 'narrower' is better than 'smaller'.
We appreciate the distinction, and have changed text accordingly.

11.10: please list the three techniques.
These are already given as the three sub-sub-headings that follow.

12.5: I guess the point to note is that the techniques are typically tuned to specific altimeters.
Thresholds on various parameters tend to be chosen separately for each instrument, but also (as noted later in the paragraph) they will be dependent upon the ground processing options that are often applied before the user sees the data.

12.11: It would be useful, in a review article, to say something about actual lead widths in this context.

New text added: "Leads range is width from a few metres to many kilometres (Lindsay & Rothrock, 1995), and, in winter, are much more frequent at the edges of the ice cover than in the centre (Willmes and Heinemann, 2016). However, the narrow ones provide a major contribution to air-sea heat fluxes (Marcq & Weiss, 2012)."

13.5: I think you mean ice drift and not currents. Also, 100 m hr-1 is not a good average number to use for the Arctic – more useful to provide a range.
We agree. In the radar section we had noted that ice drift speeds can be up to 40 km/day; that value is now used here too.

13.10: what is water-leaving radiance?
It is the inferred visible light (at different wavelengths) emanating from the surface, calculated from the satellite observations of top-of-atmosphere light levels corrected for the expected attenuation and scattering within the atmosphere. It is a technical term in ocean colour sensing, and best left unchanged in this context.

13.35: this doesn't bode well for the use of SAR imagery. Dark areas in SAR imagery are not necessarily leads (meaning open leads); they could be quite thick first-year ice.
True -- we have added "However, young ice not affected by rafting can also have a smooth flat surface and thus appear dark in SAR images" to the text, and also referred to this ambiguity in classification later on. It supports our point that validation of classification is difficult because automated processing of the reference datasets (visible or SAR) will sometimes highlight features that are not leads, leading to operator selection of "good events" for comparisons.

Section 2: Thus far, the use of other data sets to validate surface classification seems quite discouraging, wouldn't you say?
It is certainly far from simple, with the need to match up high-resolution datasets (probably allowing for ice drift) and with features in both visible and SAR imagery being open to alternative interpretations. It has been curious to note that many people illustrate the "agreement" with one or two specially selected scenes, using the overlay of data, but do not attempt to quantify the agreement across many scenes using an automatic procedure

17.2: how does the thermal noise in the altimeter translate into ranging noise?
It has little impact; the multiplicative noise has a much greater effect on range noise. The interested reader is recommended to consult Fu & Cazenave.

Figure 14. The area over which these anomalies are calculated should be shown.
That is a very good idea! Relevant boxes have been added to Fig. 1, with mention in the caption of Fig. 14, and with improved discussion in the text.

26.1: The different geophysical processes that contribute to sea surface heights should be described and why they need to be removed for estimation of dynamic ocean topography and the calculation of geostrophic current should be clarified.
Atmospheric and ocean corrections are described in sections 4.1.1 and 4.1.2 respectively. Errors in these contribute high frequency noise to the DOT (dynamic ocean topography) and

geostrophic current that are defined over longer time scales (i.e. weeks) and length scales (~100 km). Errors on the DOT are calculated at orbit crossovers (Peacock and Laxon, 2004; Giles et al., 2012) or statistically at gridded level (Dotto et al., 2018; Lawrence et al., 2018).

26.15 The distinction between DAC and inverted barometer should be described.
The inverse barometer correction (IB) only takes into account the linear effect of the atmospheric pressure on the sea level (1HPa <-> 1cm). However the response of the ocean is more complex, mainly because of the bathymetry and the winds. The DAC, also known as the high frequency response to the IB, takes into account these effects that aims to a non neglectable correction. We have added the following text:  'The latter, often termed the "inverse barometer correction" does not incorporate the delayed response caused by water requiring a finite time to move.'

Section 4.1: discussion of a number of tides is missing, e.g., solid earth, pole, etc. Even though the authors chose to discuss two that are important, the others should be mentioned. I think the magnitude of these tides in the Arctic should be listed.
We will improve the discussion of corrections (not re-written yet), noting that Ricker at al. (RSE 2016) already provides some discussion of their relative importances.

28.10: The length scale over which residual ocean tilt (due to eddies and other circulation patterns) may affect freeboard calculations is important and should be discussed.
The Rossby radius of deformation is very small at polar latitudes, so the smallest mesoscale eddies cannot be adequately resolved without a perfect fusion of data from many satellites (and even then there is only one providing coverage north of 81.5N).  However long-term changes in flows, such as the main currents or the Beaufort Gyre can be resolved.  We have not yet written text to discuss this, but will do so.

28 – The current developments in estimation of snow depth should receive more attention.
We are aware of such ongoing efforts (some of the co-authors of this paper are actively leading such projects) and have added some additional references in this section.

32.30 The authors failed to mention the potential effect of salinity on returns – due to brine wicking- from the snow-ice boundary (reported in Nandan et al., 2018). This should be discussed.
We agree that this effect and other effects of the stratification and composition of the snow layer are crucial in controlling its interaction with the radar frequencies (see for example Willat et al., 2011).

Section 6.4 seems to be outside of scope of radar altimetry.
Section 6.4 is about combining altimeter-derived estimates with those from SMOS, and thus surely relevant to a "survey" of radar altimetry techniques.

40.1: It should be snow properties, including salinity and stratigraphy, rather than just density and depth.
Agreed -- several of the authors of this paper are involved in the MOSAIC expedition and have snow-related projects to look specifically at this questions on the role of the snow

properties on the radar retrieved signal .  The relevance of other properties of the snow is now discussed in the revised paper.

Section 7: The second paragraph suggests that retracking is the primary concern, and perhaps it is for altimetry experts. But, understanding the geophysical medium being sensed, which controls the returns and therefore the geophysical retrievals, is as important if not more so. This aspect should be emphasized.
The following text has been added:  "An important aspect of this is the need to also improve the quality of the corrections, especially the density of ice and the depth and density of the overlying snow and its microstructure, some of which will be addressed by a major international in situ campaign (MOSAIC, 2018).".

**Response to Reviewer 2**

This paper presents a broad review of radar altimetry for the Arctic Ocean. While I thought many of the topics were relevant and needed for a review paper, I felt that in general the paper does not demonstrate or present a sufficient depth of understanding of the subject as there are many poorly worded statements and factual errors present throughout. Furthermore, I felt that the manuscript presented an overt subjective bias towards the authors' works rather than presenting a more objective review of the subject in general. Specific comments on these aspects are noted below.
We have corrected the factual errors pointed out to us, if we agree that they are wrong. Secondly, this paper originated from a meeting of a large number of European experts in this field, and may have had a Euro-centric bias.  Although there were still many papers from elsewhere, we have added in further relevant papers from outside Europe.  Additional papers now cited:
      Marcq and Weiss (2012)
      Willmes and Heinemann (2016)
      Chevallier et al (2017)
      Petty et al (2018)*
      Willat et al (2011)
      Proshutinsky et al (2004)
      Proshutinsky et al (2015)
      Kwok, R. and Rothrock (2009)
      Dettmerring et al (2018)*
      Dierking (2013)
      Nandan et al (2017)
      Giles et al (2008)
      Laforge et al (2018)*
      Smith & Scharroo (2015)
      Carrere et al (2016)
      GCOS (2011)
      Zygmuntowska et al (2013)

        Tonboe et al (2006)
        Nandan et al (2017)
        Xia & Xie (2018)
        Fleury et al (2017)"

of which only those asterisked include one of us as an author.

I also agree with much of what was stated in the previous review and would like to see a significant revision addressing those points as well as the ones raised here prior to publication.
See response to Reviewer 1

P2 L9-11: This statement about the current year warming of Greenland seems a bit out of context in a review paper like this.
The goal was to emphasise the huge change occuring in Arctic, and the whole Arctic environment is impacted, not only the sea.

P2 L13: State that this is an expectation of a summer ice-free Arctic Ocean, not yearround.
Indeed, corrected.

Figure 1b) Check that the grey circle area is correct, I think the SSM/I pole hole is much smaller than this.
The reviewer is correct that the "pole hole" for SSM/I should correspond to 87°N.  The dataset used for the illustration spanned both SMMR & SSM/I measurements, and thus the disk shown represents the broader disk (85°N) for SMMR observations.  The caption for Fig. 1 has been changed accordingly.

P4 L3: The sampling resolution is different than the footprint size, it would be useful to distinguish that here.
We noted that Reviewer 1 was confused about this, so have removed along-track sampling from the opening paragraph of Section 1.2, and discussed both the spacing of sampling and the resolution in the third paragraph on LRM or "pulse-limited" instruments..

P4 L17 and throughout: I think LRM may have been a term adopted more specifically for CryoSat-2, I would say "pulse limited" here instead.
We will stick with LRM, as that is the term that is now mainly used within the altimetry community  However we also introduce here the equivalent terms "conventional" and "pulse-limited".

Table 1: The antenna pattern of CryoSat-2 is non-circular, not the antenna itself.
According to https://earth.esa.int/web/eoportal/satellite-missions/c-missions/cryosat-2 "The primary super-elliptic reflectors are about 1.1 m x 1.2 m in size."

P5 L2: I'm not sure what is meant by "impulse-like shape". By definition the impulse response of an altimeter takes into account geometric spreading, so I don't think Figure 3 shows this aspect. The SAR processed waveforms do indeed have a much faster decay time than returns from only pulse-limited systems though.

We have replaced the term "more impulse-like" with "narrower", as we agree that this is clearer.

P6 L6: It's not just a smooth curve that is fitted to the return, as described in the cited references it is actually a mathematical function which takes into account the instrument impulse response, point target response, and surface roughness.
We meant a "smooth functional form", but agree that our wording was ambiguous, and has now been changed.

P6 L10: I think the use of reflectivity here is incorrect. Do you mean the reflection from the Fresnel coefficient? That is not what leads to the range of different sigma_0 values, rather it is the wide variations in surface roughness.
We have changed the word to "backscatter".

P6 L10-14: This effectively ignores why the returns from leads are confined to a small number of range bins. It also needs to be made clear that the leads represent a small spatial area which is what leads to the return being much more like that from the point target response.
We agree that simply stating that leads are "mirror-like" did not give a full explanation, so now expand that last sentence as follows "The specular reflections from such a surface mean that only points directly at nadir contribute significantly, so virtually all the power in the signal originates from a small area, and is thus confined to a small number of waveform bins."

Figure 4 b): I'm not sure this is a fair comparison between Envisat and AltiKA since the y-axis is in units of instrument counts, unless the conversion between instrument counts and power is the same between both satellites.
There is no single conversion between power and counts; the Automatic Gain Control on-board the spacecraft adjusts the relationship in a well-determined way so that the potential dynamic range is used effectively. This works well over the ocean, where rates of change are usually moderate, but the AGC on AltiKa is not optimised for the rapid changes over leads, so that the received signal exceeds the maximum permitted number of counts. We feel that the illustration and discussion are sufficient for the general reader without writing paragraphs on the AGC and deficiencies in its predictive ability.

P8 L17-18: I don't think it is true that the actual thresholds are unimportant. For example, Armitage and Davidson, 2014 found that the value of pulse peakiness used had a relationship with the bias in retrieved sea surface height.
The various parameters (such as pulse peakiness, for example) have broad continuous distributions, with some part of the range being associated with a given surface type. Although having a PP threshold of 3 or 30 does make a difference, there is only a small difference in the discrimination if the threshold is 25 or 30 -- it is simply a compromise between required confidence in the classification and the data density required for the analysis. The choice of a threshold may depend upon application, e.g. whether the purpose is study of long-term climate, or nowcasting the mesoscale currents to deal with an oil spill.

Figure 6: What is Brownian + pic?
It is a Brown-like echo with a peak; this part of the figure is now relabelled.

P10 L16: I don't believe the Doppler shift of the echoes are explicitly recorded, but are rather used within the processing loop itself. See Wingham et al., 2006.
We agree that Doppler shift is not explicitly recorded, but that the I,Q information is telemetered down enabling on-ground processing to consider time delay & Doppler cells. This is now reworded as "In SAR altimetry, the information telemetered to the ground station can be used to infer the time delay and the Doppler shift of the echoes."

P11 L8-9: I don't think this statement is true. Since even small leads have a much higher backscatter than the surrounding ice floes they tend to dominate the echoes from both SAR and traditional pulse limited systems. However, the SAR processing limits the contribution from off-nadir leads in the along track direction.
Our original text was poorly worded. By being able to minimise the effect of off-nadir leads, SAR altimeters can record the surface of ice floes nearer to leads than would LRM instruments; consequently the section of track dominated by the lead appears narrower. Thus we now state the it has "the potential to provide sea-ice observations closer to the edge of leads. Nevertheless, waveforms returned over a floe may still be strongly affected by bright reflections from off-nadir leads, especially those to the side of the track."

Figure 7: I'm unsure what this figure is showing. The method used for the threshold on the stack peakiness is unclear as no threshold is specified. Additionally it lacks context, how does it compare to previously published methods?
The Figure shows how lead-like features are seen by the statistical parameters that have been proposed in the literature to analyse the stack. We agree with the reviewer that the purpose of the classifications should be clarified. Firstly, it is important to know that both the studies, as any other classification, are based on thresholding. The caption is therefore changed into "Points selected according to classification methods based on thresholds on Stack parameters from Passaro et al. 2017 and Ricker et al. 2016 are highlighted". Fully explaining the different thresholds used in both methods is outside the scope of this paper. Following the reviewer's suggestion we complement the description with the following statement: "In the Figure, the leads identified by the classification from Ricker et al., 2016 (blue squares) and from Passaro et al., 2017 (red circles) are reported. Details of the adopted thresholds can be found in the respective studies. Despite the similarities, considerable differences remain in particular in the number of echoes classified as leads. This shows that although the statistical indicators based on RIP statistics show similar patterns, differences in selection criteria have a strong impact on classification results and it remains challenging to set appropriate thresholds that allow effective discrimination."

P12 L5: I think more importantly it is distinguishing between returns from melt ponds and returns from leads, not just recognizing melt ponds.
That is exactly what was meant; we have changed "recognition" to "discrimination", so that it is now clearer to the reviewer.

P 13 L6-7: I'm not sure what is meant here "...have been manually edited for the effects of land and clouds, with no automation of the processing."
Changed to "with operator-controlled selection of regions to avoid the effects of land and clouds, rather than full automation of the processing."

P13 second paragraph: I think this paragraph lacks a good description of what is important for radar altimetry discrimination of surface types.
The reviewer's comment is confusing. The whole of section 2 discussed details of radar altimetry discrimination. This paragraph is about the optical discrimination; the preceding paragraph had stated the validation requirements of sub-kilometre resolution and a good match-up in time.

P13 L32-33: Any smooth surface can produce a low backscatter value for a SAR in-strument, the surface could simply be very smooth first year ice and is not necessarily a lead.
True -- we have added "However, young ice not affected by rafting can also have a smooth flat surface and thus appear dark in SAR images" to the text, and also referred to this ambiguity in classification later on. It supports our point that validation of classification is difficult because automated processing of the reference datasets (visible or SAR) will sometimes highlight features that are not leads, leading to operator selection of "good events" for comparisons.

Section 2.3.2: This section focuses much on validation of surface classifications of a few select algorithms, but I am quite surprised it lacks comparison with that used by Laxon et al in various studies as these are widely employed in the standard ESA processing for CryoSat-2 and have a long history of use going back to ERS and Envisat.
Detailed comparison of the performance of each algorithms is beyond the scope of this review as it will require comparison of along-track lead detection skill (note that such along track data are often not made available from the processing centres). We have now added an additional reference to Peacock and Laxon (2004) where the authors perform a comparison of their lead detection algorithm with 'exactly coincident' ATSR-2 images.

P16 L10: I think the written out acronym here is wrong.
The reviewer is correct; the second word should be "Topographic".

P16 L13-14: These studies did not use the imagery data in a comparative sense as stated here.
Figure 5 of Kurtz et al (2013) effectively does do a qualitative comparison, but to please the reviewer text is reworded as "used in an assessment of data from the ATM."

P16 L18-25: This paragraph is poorly written and carries a negative tone towards the cited studies. The King reference is lacking a year, and I don't think the Connor et al., 2009 citation accurately represents what was shown in the study.
The year (2018) now added to King reference. The reviewer's comment about Connor lacks clarity: Figure 4 of that paper shows good alignment of features for a couple of cases, but there is no quantitative (statistical) comparison of the classification, so we cannot cite

measures of how effective it is. If the reviewer feels that some other point from Connor study should be brought out, they need to be more specific. Text has been rephrased to cite ASIRAS classification by Zygmuntowska et al. (2013) and provide a recommendation: "It is hoped that future studies will investigate comparing waveform classification from ASIRAS and spaceborne altimeters."

P16 L31-32: This is a poorly written sentence.
First part of sentence rewritten as "The received radar signal (waveform) is the sum of reflections from multiple facets within the instrument footprint"

P17 L9-11: This statement is not true, the returns from sea ice are quite a bit different than open ocean returns most prominently they tend to have a much shorter trailing edge, but this is not due to snow volume scattering.
We agree with the reviewer that there are differences between ocean and sea ice waveforms, especially regarding the trailing edge. Nevertheless, there are also some similarities, as shown in Figure 3. Text rewritten as:
"There are similarities in the waveforms from unbroken sea ice and from ocean, but the former has more variability in the trailing edge (see Figure 3). This is because the sea-ice surface differs from open ocean by its snow cover, which may contribute to the waveform shape, as well as by the surface roughness distribution. On sea ice, the roughness is constituted from different components (e.g., snow-covered level ice or blocky ridges) as well as internal scatterers, compared with the ocean surface that is a single physical medium. Single waveform shape parameters, such as the backscatter coefficient might have overlapping ranges for rough sea ice and open ocean, but the two surface types can usually be easily discriminated using a sea ice concentration mask. Consequently, some approaches for determining range to ice floes have a strong inheritance of the physical retrackers used over the ocean, although others are more empirical."

P17 L17-19: The Giles et al., 2007 paper did not use an OCOG retracker. I think in general this section is missing much of the historical work done by Seymour Laxon in retracking of floes.
The reference should have been to Giles et al 2008, not Giles et al 2007. Giles et al 2008 did use the same retracker for floes as Laxon 2003. Although we are not attempting to write a historical account of altimetry, we do agree that some important contributions from Seymour Laxon have been omitted. Indeed, the base for retracking ice floes in the first place was built in UCL about two decades ago, and the revised version of this paper will more clearly reflect his pioneering work. (Given that one of our authors is from UCL/CPOM, this cannot be an institutional bias!).

P18 L1-7: This section clearly misrepresents the historical work done towards development of CryoSat-2 retracking. The widely cited Laxon et al., 2013 study utilized a threshold tracker for sea ice floes and did not use an OCOG tracker. This threshold tracker superseded the TFMRA tracker which is quite similar on a conceptual level.
Opening text of that subsection now reads: "
Many of the published CryoSat-2 sea ice thickness studies, including the first one by Laxon et al 2013, use an empirical retracker for sea-ice floe echoes. The main advantage of these

is that they are simple and easy to realise, yet match well to independent validation data. Laxon et al. 2013 introduced a retracking scheme where the retracked point is set at 70% of the first local maximum power."

P18 L 9-11: I'm not sure what this has to do with how snow impacts the results?
We agree that the original paragraph was slightly misleading. Snow affects the leading edge by its properties. For example, wet snow will shift the main reflecting horizon towards the snow surface. Moreover, scattering from internal layers (ice lenses) may also alter the reflecting horizon. On the other hand, surface roughness plays an important role here as well. All these characteristics vary in space and time, and ideally this needs to be considered in the retracking algorithm. We have revised this paragraph for clarification.

P18 L13-14: I think it is quite a subjective statement on the author's part that TFMRA is the most widely used tracker for SAR data and should be removed. I think the statement also does a disservice to the historical development of sea ice retrackers in general by framing this in such a way, and conveys a bias towards the authors' own work.
We agree that the original text in the manuscript is too easily interpreted as just advertising TFMRA. The revised version gives Laxon 2013 and other retracking schemes more credit. However, of currently available Cryosat-2 sea ice thickness products the majority do use a threshold retracking scheme - the notable exception being the two NASA products.

Section 3.1.3: I find this section to be poorly written. The second paragraph inappropriately mixes in a discussion of models for ice and open ocean returns even though it has been quite clearly shown in prior studies that the models for open ocean returns do not work for sea ice regions. The section also ignores what physical modeling of ice returns have shown with describing variations in the radar returns as elaborated on in previous sections.
This section discusses the idea of retrackers that fit an expected shape model to the data, as opposed to those that simply look at a threshold, for instance. The heritage is from ocean retrackers, as in that case the surface is more readily modelled, but the section quite clearly progresses to talk about ideas for implementation over ice floes by adapting the surface scattering model.

P18 L30: The Rivas et al., 2006 paper did not compare sea ice and open ocean roughness so this sentence needs some revision.
Reference removed, as it was erroneously cited.

P19 L9-10: How is the waveform "severely undersampled" for sea ice leads? The range resolution sampling is set by the bandwidth of the altimeter. So it would seem to me more appropriate to discuss this in the context of the need for high accuracy or precision in the data which is limited by the bandwidth of the satellite altimeters.
The phrase "severely undersampled" has been replaced by "poorly resolved" and accompanied by a discussion of the bandwidth issues. "Jenson (1999) showed that the process of power detection (i.e. squaring the signal) effectively changes the bandwidth of the processor and thus its resolution; Smith & Scharroo (2015) advocate the application of zero-padding prior to the Fourier Transform to prevent this loss in information."

P19 L11-15: Here I think it is missed that the work of Laxon et al., 2013 used the tracker from Giles et al., 2007 and this is used in the CryoSat-2 sea ice product. Again, this paragraph conveys a bias towards the authors' own work.
Laxon 2013 and Giles 2007 retracking scheme will be discussed in more detail in the revised manuscript. It is novel to see a reviewer requesting that we cite more often the papers from one of our own institutions!

P19 L20-26: This paragraph ignores the fact that the range resolution is determined by the received bandwidth, zero padding of the waveform does not actually confer a real increase to the range resolution, and associating this to the calibration functions used to obtain mm level path delays is highly inappropriate. Zero padding can be useful in certain algorithm approaches or in visualizing the data, but this needs to be more clearly written.
"finer resolution" changed to "finer sampling", and the benefits of zero padding briefly explained by "This process, which is most relevant for very sharp leading edges, avoids the loss of information that otherwise occurs upon squaring the voltages (Smith & Scharroo, 2015)"

P19 L29-30: Here I think the authors miss an important need for zero padding of SAR waveforms as demonstrated by the cited Jenson, 1999 study.
Jenson paper now cited here.

Section 3.3: The cited study by Dinardo et al., 2017 did not demonstrate retrievals over sea ice or leads, but was focused on the coasts of German Bight and West Baltic Sea.
So it seems not credible to put the claim in this review paper that the methodology was successfully demonstrated to retrieve sea ice freeboard from CryoSat-2 SAR data. I think here the authors also miss discussing the Kurtz et al., 2014 study which showed a unified physical tracker for both sea ice leads and floes from SAR waveforms. Again, this seems to show clear biases towards the authors' own work.
A more appropriate reference for the adaption of the SAMOSA model has now been used (Fleury et al 2017, Laforge et al., 2018), and we now also discuss the Kurtz et al (2014) retracker, whose previous omission was a great mistake as it was the first study using a physical retracker over sea-ice.

Section 3.4.1: Here the authors miss the opportunity to more thoroughly describe the capability of CryoSat-2 to determine phase and thus estimate the range bias present over sea ice as shown in Armitage and Davidson.
SARIN mode can help mitigate the bias from off-nadir lead bias or snagging as demonstrated in Armitage and Davidson (2014), which we now cite here. Note, we already had a discussion of SARin in section 6.3.

P23 L11: This sentence is not accurate: "This is due to the along-track beam-limited resolution, which also reduces the backscatter from across-track points." The SAR processing does not impact the backscatter from across-track points in this way.
After some internal discussion, we agree with the reviewer, and have changed the text to read "but the response to across-track bright targets (such as leads) remains the same."

P25 L4: The western Greenland Sea is quite often ice-covered.
The statement should have referred to the specific part of the Greenland Sea being analysed, also with the caveat that a "mean sea level" can be recovered if just some part of that box is ice-free.  This has now been reworded as: "Whilst the selected part of the Greenland Sea is rarely fully ice-covered,".

Section 4.1.1: Here it would be useful to state the magnitude and variability of the corrections.
We agree. Some of the amplitudes and variabilities for the given models in the CryoSat-2 products are given in Ricker et al. (2016); we are still considering how to include similar in the revised version of this paper.

P29 L8-9: Not all studies use a 50% reduction for snow climatology, the Kwok and Cunningham study used a different value.
Text has been modified, in light of the reviewer's comment:
"Most snow climatology-based sea-ice thickness studies apply a 50% snow depth reduction for areas of FYI. The notable exception is Kwok and Cunningham (2015), who found that the agreement between validation data and CS-2 based thickness improved when using only 30% reduction for snow on FYI."
.
Section 4.2.2: I think here the important factor of snow depth and density in leading to a lower propagation speed of light in the snow is missed.
This will be added in the revised manuscript.

P32 L18, P34 L4-5: From the text it seems unclear what is measured by Operation Ice-Bridge, in my reading it seems to imply only laser altimeter freeboard. But it should be noted that snow depth measurements from radar altimetry are also used to determine snow.
Text modified at 4 places (given by old page/line numbering) to address the snow depth measurements too.
i) p32 L18 has been changed to: "One is from NASA's Operation IceBridge, with the sea ice freeboard given by the total freeboard (snow + sea ice) measured by laser altimeter minus the snow depth measured by the snow radar".
ii) p32 L23 "The measurements obtained by the OIB snow radar also provide valuable validation of snow depth".
iii) p33 l10 has been changed to "..., total (sea ice + snow) thickness by airborne electromagnetic ... "
iv) P34 l1 has been changed to ", or total thickness ..."
v) P34 l3-4 has been changed to: "In addition, the sea ice thickness obtained by OIB are derived by the measured total freeboard and snow depth according to Eq. 2."

**Response to Reviewer 3**

The paper 'Review of Radar Altimetry Techniques over the Arctic Ocean: Recent Progress and Future Opportunities for Sea Level and Sea Ice Research by Quartly et al. aim to provide an overview and review of various radar altimetry techniques over the Arctic Ocean for sea ice freeboard/thickness and sea surface height retrievals. The paper reviews altimetry techniques in a very broad context and includes a comprehensive review of altimetric data sets, basic-to-advanced concepts of radar altimetry, retracking algorithms, and geophysical correction methods for unambiguous parameter retrievals.

Since I am an altimetry expert on sea ice freeboard/thickness retrievals, I would want to concentrate my focus on reviewing the sea ice side of this paper. Overall, I find this manuscript not suitable for publication at this point of time, as the authors fail to provide a comprehensive treatment snow depth and its associated thermal and geophysical properties as a major source of uncertainty in sea ice freeboard/thickness retrievals. The authors should understand the impact of critical snow geophysical properties such as snow temperature, snow salinity, snow grain microstructure, in addition to snow thickness and density (which the authors have already pointed out as error sources in this manuscript). We recognise that the absence of discussion of the effect of various snow properties was a major omission, and are adding text to make amends.

This is to be emphasized in more detail as snow covers on sea ice is still considered to be the greatest source of uncertainty, with currently used snow thickness and density assumptions considered to be unassumingly vague. A critical assessment of how snow covers and its geophysical properties impact sea ice thickness freeboard/thickness should be accounted for in this manuscript and requires substantial focus in this manuscript. I would recommend you reviewing latest literature on how snow thickness (Tonboe et al., 2006; Tonboe et al., 2010; Ricker et al., 2014, 2015, 2017), snow density (Alexandrov et al., 2010, Tonboe et al., 2006), snow temperature (Willatt et al., 2011), snow compression (Tilling et al., 2015; Kern et al., 2015), snow salinity (Nandan et al., 2017; Nandan et al., 2016) impact freeboard retrievals from radar altimetry.
We agree with the reviewer that the effects of snow temperature, snow grain microstructure density and salinity are not well represented in these sections. We have therefore added a few paragraphs discussing this in more detail.

I also agree with the comments from the other two reviewers and would like to see a more detailed revision addressing the points above and also from the other reviewers. Hence, I do not recommend publication in its present stage, and would recommend to resubmit this manuscript in a revised form. I will be happy to provide specific comments to the revised manuscript, once the general comments are addressed.

**Author-instigated changes**

Bandwidth of AltiKa is now "480 MHz"
Reference 'Bella, A. D." corrected to "Di Bella, A.".

Now acknowledges NSIDC DAAC rather than NSIDC.